# Sparse and Continuous Attention Mechanisms

**André F. T. Martins**[,Ω,♭,℮]  **António Farinhas**[,Ω]  **Marcos Treviso**[,Ω]

**Vlad Niculae**[ℳ,,Ω]  **Pedro M. Q. Aguiar**[♮,♭]  **Mário A. T. Figueiredo**[,Ω,♭]

{andre.t.martins, marcos.treviso, antonio.farinhas, mario.figueiredo}
@tecnico.ulisboa.pt, aguiar@isr.ist.utl.pt, vlad@vene.ro

[Ω]Instituto de Telecomunicações, Instituto Superior Técnico, Lisbon, Portugal
[♮]Instituto de Sistemas e Robótica, Instituto Superior Técnico, Lisbon, Portugal
[♭]LUMLIS (Lisbon ELLIS Unit), Lisbon, Portugal
[ℳ]Informatics Institute, University of Amsterdam, The Netherlands
[℮]Unbabel, Lisbon, Portugal

## Abstract

Exponential families are widely used in machine learning; they include many distributions in continuous and discrete domains (*e.g.*, Gaussian, Dirichlet, Poisson, and categorical distributions via the softmax transformation). Distributions in each of these families have fixed support. In contrast, for finite domains, there has been recent work on sparse alternatives to softmax (*e.g.* sparsemax and $\alpha$-entmax), which have varying support, being able to assign zero probability to irrelevant categories. This paper expands that work in two directions: first, we extend $\alpha$-entmax to continuous domains, revealing a link with Tsallis statistics and deformed exponential families. Second, we introduce continuous-domain attention mechanisms, deriving efficient gradient backpropagation algorithms for $\alpha \in \{1, 2\}$. Experiments on attention-based text classification, machine translation, and visual question answering illustrate the use of continuous attention in 1D and 2D, showing that it allows attending to time intervals and compact regions.

## 1 Introduction

Exponential families are ubiquitous in statistics and machine learning [1, 2]. They enjoy many useful properties, such as the existence of conjugate priors (crucial in Bayesian inference) and the classical Pitman-Koopman-Darmois theorem [3–5], which states that, among families with **fixed support** (independent of the parameters), exponential families are the only having sufficient statistics of fixed dimension for any number of i.i.d. samples.

Departing from exponential families, there has been recent work on discrete, finite-domain distributions with **varying and sparse support**, via the *sparsemax* and the *entmax* transformations [6–8]. Those approaches drop the link to exponential families of categorical distributions provided by the softmax transformation, which always yields dense probability mass functions. In contrast, sparsemax and entmax can lead to sparse distributions, whose support is not constant throughout the family. This property has been used to design sparse attention mechanisms with improved interpretability [8, 9].

However, sparsemax and entmax are so far limited to discrete domains. Can a similar approach be extended to continuous domains? This paper provides that extension and pinpoints a connection with "deformed exponential families" [10–12] and Tsallis statistics [13], leading to $\alpha$**-sparse families** (§2).

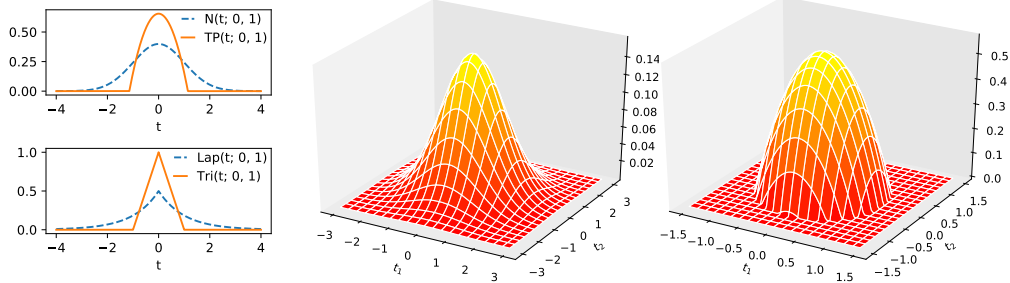

Figure 1: **1D and 2D distributions generated by the $\Omega_\alpha$-RPM for $\alpha \in \{1, 2\}$.** Left: Univariate location-scale families, including Gaussian and truncated parabola (top) and Laplace and triangular (bottom). Middle and right: Bivariate Gaussian $\mathcal{N}(t; 0, I)$ and truncated paraboloid $\mathrm{TP}(t; 0, I)$.

We use this construction to obtain new density families with varying support, including the *truncated parabola* and *paraboloid* distributions (2-sparse counterpart of the Gaussian, §2.4 and Fig. 1).

Softmax and its variants are widely used in *attention mechanisms*, an important component of neural networks [14]. Attention-based neural networks can "attend" to finite sets of objects and identify relevant features. We use our extension above to devise new **continuous attention mechanisms** (§3), which can attend to continuous data streams and to domains that are inherently continuous, such as images. Unlike traditional attention mechanisms, ours are suitable for selecting compact regions, such as 1D-segments or 2D-ellipses. We show that the Jacobian of these transformations are generalized covariances, and we use this fact to obtain efficient backpropagation algorithms (§3.2).

As a proof of concept, we apply our models with continuous attention to text classification, machine translation, and visual question answering tasks, with encouraging results (§4).

**Notation.** Let $(S, \mathcal{A}, \nu)$ be a measure space, where $S$ is a set, $\mathcal{A}$ is a $\sigma$-algebra, and $\nu$ is a measure. We denote by $\mathcal{M}_+^1(S)$ the set of $\nu$-absolutely continuous probability measures. From the Radon-Nikodym theorem [15, §31], each element of $\mathcal{M}_+^1(S)$ is identified (up to equivalence within measure zero) with a probability density function $p : S \to \mathbb{R}_+$, with $\int_S p(t) \, d\nu(t) = 1$. For convenience, we often drop $d\nu(t)$ from the integral. We denote the measure of $A \in \mathcal{A}$ as $|A| = \nu(A) = \int_A 1$, and the support of a density $p \in \mathcal{M}_+^1(S)$ as $\mathrm{supp}(p) = \{t \in S \mid p(t) > 0\}$. Given $\phi : S \to \mathbb{R}^m$, we write expectations as $\mathbb{E}_p[\phi(t)] := \int_S p(t) \, \phi(t)$. Finally, we define $[a]_+ := \max\{a, 0\}$.

## 2 Sparse Families

In this section, we provide background on exponential families and its generalization through Tsallis statistics. We link these concepts, studied in statistical physics, to sparse alternatives to softmax recently proposed in the machine learning literature [6, 8], extending the latter to continuous domains.

### 2.1 Regularized prediction maps ($\Omega$-RPM)

Our starting point is the notion of $\Omega$-regularized prediction maps, introduced by Blondel et al. [7] for finite domains $S$. This is a general framework for mapping vectors in $\mathbb{R}^{|S|}$ (*e.g.*, label scores computed by a neural network) into probability vectors in $\triangle^{|S|}$ (the simplex), with a regularizer $\Omega$ encouraging uniform distributions. Particular choices of $\Omega$ recover argmax, softmax [16], and sparsemax [6]. Our definition below extends this framework to arbitrary measure spaces $\mathcal{M}_+^1(S)$, where we assume $\Omega : \mathcal{M}_+^1(S) \to \mathbb{R}$ is a lower semi-continuous, proper, and strictly convex function.

**Definition 1.** *The $\Omega$-regularized prediction map ($\Omega$-RPM) $\hat{p}_\Omega : \mathcal{F} \to \mathcal{M}_+^1(S)$ is defined as*

$$\hat{p}_\Omega[f] = \arg\max_{p \in \mathcal{M}_+^1(S)} \mathbb{E}_p[f(t)] - \Omega(p), \tag{1}$$

*where $\mathcal{F}$ is the set of functions for which the maximizer above exists and is unique.*

It is often convenient to consider a "temperature parameter" $\tau > 0$, absorbed into $\Omega$ via $\Omega := \tau\tilde{\Omega}$. If $f$ has a unique global maximizer $t^\star$, the low-temperature limit yields $\lim_{\tau\to 0} \hat{p}_{\tau\tilde{\Omega}}[f] = \delta_{t^\star}$, a Dirac delta distribution at the maximizer of $f$. For finite $S$, this is the *argmax* transformation shown in [7]. Other interesting examples of regularization functionals are shown in the next subsections.

## 2.2 Shannon's negentropy and exponential families

A natural choice of regularizer is the Shannon's negentropy, $\Omega(p) = \int_S p(t)\log p(t)$. In this case, if we interpret $-f(t)$ as an energy function, the $\Omega$-RPM corresponds to the well-known *free energy variational principle*, leading to Boltzmann-Gibbs distributions ([17]; see App. A):

$$\hat{p}_\Omega[f](t) = \frac{\exp(f(t))}{\int_S \exp(f(t'))d\nu(t')} = \exp\big(f(t) - A(f)\big), \tag{2}$$

where $A(f) := \log \int_S \exp(f(t))$ is the log-partition function. If $S$ is finite and $\nu$ is the counting measure, the integral in (2) is a summation and we can write $f$ as a vector $[f_1, \ldots, f_{|S|}] \in \mathbb{R}^{|S|}$. In this case, the $\Omega$-RPM is the *softmax transformation*,

$$\hat{p}_\Omega[f] = \mathrm{softmax}(f) = \frac{\exp(f)}{\sum_{k=1}^{|S|} \exp(f_k)} \in \triangle^{|S|}. \tag{3}$$

If $S = \mathbb{R}^N$, $\nu$ is the Lebesgue measure, and $f(t) = -^1\!/_2(t-\mu)^\top \Sigma^{-1}(t-\mu)$ for $\mu \in \mathbb{R}^N$ and $\Sigma \succ 0$ (i.e., $\Sigma$ is a positive definite matrix), we obtain a *multivariate Gaussian*, $\hat{p}_\Omega[f](t) = \mathcal{N}(t; \mu, \Sigma)$. This becomes a univariate Gaussian $\mathcal{N}(t; \mu, \sigma^2)$ if $N = 1$. For $S = \mathbb{R}$ and defining $f(t) = -|t - \mu|/b$, with $\mu \in \mathbb{R}$ and $b > 0$, we get a *Laplace* density, $\hat{p}_\Omega[f](t) = \frac{1}{2b}\exp\left(-|t-\mu|/b\right)$.

**Exponential families.** Let $f_\theta(t) = \theta^\top \phi(t)$, where $\phi(t) \in \mathbb{R}^M$ is a vector of *statistics* and $\theta \in \Theta \subseteq \mathbb{R}^M$ is a vector of *canonical parameters*. A family of the form (2) parametrized by $\theta \in \Theta \subseteq \mathbb{R}^M$ is called an *exponential family* [2]. Exponential families have many appealing properties, such as the existence of conjugate priors and sufficient statistics, and a dually flat geometric structure [18]. Many well-known distributions are exponential families, including the categorical and Gaussian distributions above, and Laplace distributions with a fixed $\mu$. A key property of exponential families is that the support is constant within the same family and dictated by the base measure $\nu$: this follows immediately from the positiveness of the $\exp$ function in (2). We abandon this property in the sequel.

## 2.3 Tsallis' entropies and $\alpha$-sparse families

Motivated by applications in statistical physics, Tsallis [13] proposed a generalization of Shannon's negentropy. This generalization is rooted on the notions of $\beta$-logarithm, $\log_\beta : \mathbb{R}_{\geq 0} \to \mathbb{R}$ (not to be confused with base-$\beta$ logarithm), and $\beta$-exponential, $\exp_\beta : \mathbb{R} \to \mathbb{R}$:

$$\log_\beta(u) := \begin{cases} \frac{u^{1-\beta}-1}{1-\beta}, & \beta \neq 1 \\ \log u, & \beta = 1; \end{cases} \qquad \exp_\beta(u) := \begin{cases} [1 + (1-\beta)u]_+^{1/(1-\beta)}, & \beta \neq 1 \\ \exp u, & \beta = 1. \end{cases} \tag{4}$$

Note that $\lim_{\beta\to 1}\log_\beta(u) = \log u$, $\lim_{\beta\to 1}\exp_\beta(u) = \exp u$, and $\log_\beta(\exp_\beta(u)) = u$ for any $\beta$. Another important concept is that of "$\beta$-escort distribution" [13]: this is the distribution $\tilde{p}^\beta$ given by

$$\tilde{p}^\beta(t) := \frac{p(t)^\beta}{\|p\|_\beta^\beta}, \quad \text{where } \|p\|_\beta^\beta = \int_S p(t')^\beta d\nu(t'). \tag{5}$$

Note that we have $\tilde{p}^1(t) = p(t)$.

The $\alpha$**-Tsallis negentropy** [19, 13] is defined as:[1]

$$\Omega_\alpha(p) := \frac{1}{\alpha}\mathbb{E}_p[\log_{2-\alpha}(p(t))] = \begin{cases} \frac{1}{\alpha(\alpha-1)}\left(\int_S p(t)^\alpha - 1\right), & \alpha \neq 1, \\ \int_S p(t)\log p(t), & \alpha = 1. \end{cases} \tag{6}$$

Note that $\lim_{\alpha\to 1}\Omega_\alpha(p) = \Omega_1(p)$, for any $p \in \mathcal{M}_+^1(S)$, with $\Omega_1(p)$ recovering Shannon's negentropy (proof in App. B). Another notable case is $\Omega_2(p) = ^1\!/_2\int_S p(t)^2 - ^1\!/_2$, the negative of which is called the Gini-Simpson index [20, 21]. We come back to the $\alpha = 2$ case in §2.4.

For $\alpha > 0$, $\Omega_\alpha$ is strictly convex, hence it can be plugged in as the regularizer in Def. 1. The next proposition ([10]; proof in App. B) provides an expression for $\Omega_\alpha$-RPM using the $\beta$-exponential (4):

**Proposition 1.** *For $\alpha > 0$ and $f \in \mathcal{F}$,*

$$\hat{p}_{\Omega_\alpha}[f](t) = \exp_{2-\alpha}(f(t) - A_\alpha(f)), \tag{7}$$

*where $A_\alpha : \mathcal{F} \to \mathbb{R}$ is a normalizing function:* $A_\alpha(f) = \frac{\frac{1}{1-\alpha} + \int_S p_\theta(t)^{2-\alpha} f(t)}{\int_S p_\theta(t)^{2-\alpha}} - \frac{1}{1-\alpha}$.

Let us contrast (7) with Boltzmann-Gibbs distributions (2), recovered with $\alpha = 1$. One key thing to note is that the $(2 - \alpha)$-exponential, for $\alpha > 1$, can return zero values. Therefore, the distribution $\hat{p}_{\Omega_\alpha}[f]$ in (7) may not have full support, *i.e.*, we may have $\mathrm{supp}(\hat{p}_{\Omega_\alpha}[f]) \subsetneq S$. We say that $\hat{p}_{\Omega_\alpha}[f]$ has *sparse support* if $\nu(S \setminus \mathrm{supp}(\hat{p}_{\Omega_\alpha}[f])) > 0$.[2] This generalizes the notion of sparse vectors.

**Relation to sparsemax and entmax.** Blondel et al. [7] showed that, for finite $S$, $\Omega_2$-RPM is the **sparsemax** transformation, $\hat{p}_\Omega[f] = \mathrm{sparsemax}(f) = \arg\min_{p \in \triangle^{|S|}} \|p - f\|^2$. Other values of $\alpha$ were studied by Peters et al. [8], under the name $\alpha$-**entmax** transformation. For $\alpha > 1$, these transformations have a propensity for returning sparse distributions, where several entries have zero probability. Proposition 1 shows that similar properties can be obtained when $S$ is continuous.

**Deformed exponential families.** With a linear parametrization $f_\theta(t) = \theta^\top \phi(t)$, distributions with the form (7) are called *deformed exponential* or *q-exponential families* [10–12, 24]. The geometry of these families induced by the Tsallis $q$-entropy was studied by Amari [18, §4.3].[3] Unlike those prior works, we are interested in the sparse, light tail scenario ($\alpha > 1$), not in heavy tails. For $\alpha > 1$, we call these $\alpha$-**sparse families.** When $\alpha \to 1$, $\alpha$-sparse families become exponential families and they cease to be "sparse", in the sense that all distributions in the same family have the same support.

A relevant problem is that of characterizing $A_\alpha(\theta)$. When $\alpha = 1$, $A_1(\theta) = \lim_{\alpha \to 1} A_\alpha(\theta) = \log \int_S \exp(\theta^\top \phi(t))$ is the log-partition function (see (2)), and its first and higher order derivatives are equal to the moments of the sufficient statistics. The following proposition (stated as Amari and Ohara [25, Theorem 5], and proved in our App. D) characterizes $A_\alpha(\theta)$ for $\alpha \neq 1$ in terms of an expectation under the $\beta$-escort distribution for $\beta = 2 - \alpha$ (see (5)). This proposition will be used later to derive the Jacobian of entmax attention mechanisms.

**Proposition 2.** $A_\alpha(\theta)$ *is a convex function and its gradient is given by*

$$\nabla_\theta A_\alpha(\theta) = \mathbb{E}_{\tilde{p}_\theta^{2-\alpha}}[\phi(t)] = \frac{\int_S p_\theta(t)^{2-\alpha} \phi(t)}{\int_S p_\theta(t)^{2-\alpha}}. \tag{8}$$

### 2.4 The 2-Tsallis entropy: sparsemax

In this paper, we focus on the case $\alpha = 2$. For finite $S$, this corresponds to the sparsemax transformation proposed by Martins and Astudillo [6], which has appealing theoretical and computational properties. In the general case, plugging $\alpha = 2$ in (7) leads to the $\Omega_2$-RPM,

$$\hat{p}_{\Omega_2}[f](t) = [f(t) - \lambda]_+, \qquad \text{where } \lambda = A_2(f) - 1, \tag{9}$$

*i.e.*, $\hat{p}_{\Omega_2}[f]$ is obtained from $f$ by subtracting a constant (which may be negative) and truncating, where that constant $\lambda$ must be such that $\int_S [f(t) - \lambda]_+ = 1$.

If $S$ is continuous and $\nu$ the Lebesgue measure, we call $\Omega_2$-RPM the **continuous sparsemax** transformation. Examples follow, some of which correspond to novel distributions.

**Truncated parabola.** If $f(t) = -\frac{(t-\mu)^2}{2\sigma^2}$, we obtain the continuous sparsemax counterpart of a Gaussian, which we dub a "truncated parabola":

$$\hat{p}_{\Omega_2}[f](t) = \left[ -\frac{(t-\mu)^2}{2\sigma^2} - \lambda \right]_+ =: \mathrm{TP}(t; \mu, \sigma^2), \tag{10}$$

where $\lambda = -\frac{1}{2}\big(3/(2\sigma)\big)^{2/3}$ (see App. E.1). This function, depicted in Fig. 1 (top left), is widely used in density estimation. For $\mu = 0$ and $\sigma = \sqrt{2/3}$, it is known as the Epanechnikov kernel [26].

**Truncated paraboloid.** The previous example can be generalized to $S = \mathbb{R}^N$, with $f(t) = -\frac{1}{2}(t-\mu)^\top \Sigma^{-1}(t-\mu)$, where $\Sigma \succ 0$, leading to a "multivariate truncated paraboloid," the sparsemax counterpart of the multivariate Gaussian (see middle and rightmost plots in Fig. 1):

$$\hat{p}_{\Omega_2}[f](t) = \left[ -\lambda - \tfrac{1}{2}(t-\mu)\Sigma^{-1}(t-\mu) \right]_+, \qquad \text{where } \lambda = -\Big( \Gamma\big(\tfrac{N}{2}+2\big) / \sqrt{\det(2\pi\Sigma)} \Big)^{\frac{2}{2+N}}. \tag{11}$$

The expression above, derived in App. E.2, reduces to (10) for $N = 1$. Notice that (unlike in the Gaussian case) a diagonal $\Sigma$ does not lead to a product of independent truncated parabolas.

**Triangular.** Setting $f(t) = -|t - \mu|/b$, with $b > 0$, yields the triangular distribution

$$\hat{p}_{\Omega_2}[f](t) = \left[ -\lambda - \frac{|t-\mu|}{b} \right]_+ =: \mathrm{Tri}(t; \mu, b), \tag{12}$$

where $\lambda = -1/\sqrt{b}$ (see App. E.3). Fig. 1 (bottom left) depicts this distribution alongside Laplace.

**Location-scale families.** More generally, let $f_{\mu,\sigma}(t) := -\frac{1}{\sigma}g'(|t-\mu|/\sigma)$ for a location $\mu \in \mathbb{R}$ and a scale $\sigma > 0$, where $g : \mathbb{R}_+ \to \mathbb{R}$ is convex and continuously differentiable. Then, we have

$$\hat{p}_{\Omega_2}[f](t) = \left[ -\lambda - \tfrac{1}{\sigma}g'(|t-\mu|/\sigma) \right]_+, \tag{13}$$

where $\lambda = -g'(a)/\sigma$ and $a$ is the solution of the equation $ag'(a) - g(a) + g(0) = \frac{1}{2}$ (a sufficient condition for such solution to exist is $g$ being strongly convex; see App. E.4 for a proof). This example subsumes the truncated parabola ($g(t) = t^3/6$) and the triangular distribution ($g(t) = t^2/2$).

**2-sparse families.** Truncated parabola and paraboloid distributions form a 2-sparse family, with statistics $\phi(t) = [t, \mathrm{vec}(tt^\top)]$ and canonical parameters $\theta = [\Sigma^{-1}\mu, \mathrm{vec}(-\frac{1}{2}\Sigma^{-1})]$. Gaussian distributions form an exponential family with the same sufficient statistics and canonical parameters. In 1D, truncated parabola and Gaussians are both particular cases of the so-called "$q$-Gaussian" [10, §4.1], for $q = 2 - \alpha$. Triangular distributions with a fixed location $\mu$ and varying scale $b$ also form a 2-sparse family (similarly to Laplace distributions with fixed location being exponential families).

## 3 Continuous Attention

Attention mechanisms have become a key component of neural networks [14, 27, 28]. They dynamically detect and extract relevant input features (such as words in a text or regions of an image). So far, attention has only been applied to discrete domains; we generalize it to *continuous* spaces.

**Discrete attention.** Assume an input object split in $L = |S|$ pieces, *e.g.*, a document with $L$ words or an image with $L$ regions. A vanilla attention mechanism works as follows: each piece has a $D$-dimensional representation (*e.g.*, coming from an RNN or a CNN), yielding a matrix $V \in \mathbb{R}^{D \times L}$. These representations are compared against a query vector (*e.g.*, using an additive model [14]), leading to a score vector $f = [f_1, \ldots, f_L] \in \mathbb{R}^L$. Intuitively, the relevant pieces that need attention should be assigned high scores. Then, a transformation $\rho : \mathbb{R}^L \to \triangle^L$ (*e.g.*, softmax or sparsemax) is applied to the score vector to produce a probability vector $p = \rho(f)$. We may see this as an $\Omega$-RPM. The probability vector $p$ is then used to compute a weighted average of the input representations, via $c = Vp \in \mathbb{R}^D$. This context vector $c$ is finally used to produce the network's decision.

To learn via the backpropagation algorithm, the Jacobian of the transformation $\rho$, $J_\rho \in \mathbb{R}^{L \times L}$, is needed. Martins and Astudillo [6] gave expressions for softmax and sparsemax,

$$J_{\mathrm{softmax}}(f) = \mathrm{Diag}(p) - pp^\top, \qquad J_{\mathrm{sparsemax}}(f) = \mathrm{Diag}(s) - ss^\top/(1^\top s), \tag{14}$$

where $p = \mathrm{softmax}(f)$, and $s$ is a binary vector whose $\ell^{\text{th}}$ entry is 1 iff $\ell \in \mathrm{supp}(\mathrm{sparsemax}(f))$.

**Algorithm 1:** Continuous softmax attention with $S = \mathbb{R}^D$, $\Omega = \Omega_1$, and Gaussian RBFs.

---

**Parameters:** Gaussian RBFs $\psi(t) = [\mathcal{N}(t; \mu_j, \Sigma_j)]_{j=1}^N$, basis functions $\phi(t) = [t, \mathrm{vec}(tt^\top)]$, value function $V_B(t) = B\psi(t)$ with $B \in \mathbb{R}^{D \times N}$, score function $f_\theta(t) = \theta^\top \phi(t)$ with $\theta \in \mathbb{R}^M$

**Function** Forward($\theta := [\Sigma^{-1}\mu, -\frac{1}{2}\Sigma^{-1}]$)**:**
  $r_j \leftarrow \mathbb{E}_{\hat{p}_\Omega[f_\theta]}[\psi_j(t)] = \mathcal{N}(\mu, \mu_j, \Sigma + \Sigma_j), \quad \forall j \in [N]$       // Eqs. 15, 46
  **return** $c \leftarrow Br$ *(context vector)*

**Function** Backward($\frac{\partial \mathcal{L}}{\partial c}, \theta := [\Sigma^{-1}\mu, -\frac{1}{2}\Sigma^{-1}]$)**:**
  **for** $j \leftarrow 1$ **to** $N$ **do**
    $\tilde{s} \leftarrow \mathcal{N}(\mu, \mu_j, \Sigma + \Sigma_j)$, $\tilde{\Sigma} \leftarrow (\Sigma^{-1} + \Sigma_j^{-1})^{-1}$, $\tilde{\mu} \leftarrow \tilde{\Sigma}(\Sigma^{-1}\mu + \Sigma_j^{-1}\mu_j)$
    $\frac{\partial r_j}{\partial \theta} \leftarrow \mathrm{cov}_{\hat{p}_\Omega[f_\theta]}(\phi(t), \psi_j(t)) = [\tilde{s}(\tilde{\mu} - \mu); \tilde{s}(\tilde{\Sigma} + \tilde{\mu}\tilde{\mu}^\top - \Sigma - \mu\mu^\top)]$     // Eqs. 18, 47-48
  **return** $\frac{\partial \mathcal{L}}{\partial \theta} \leftarrow (\frac{\partial r}{\partial \theta})^\top B^\top \frac{\partial \mathcal{L}}{\partial c}$

---

### 3.1 The continuous case: score and value functions

Our extension of $\Omega$-RPMs to arbitrary domains (Def. 1) opens the door for constructing **continuous attention mechanisms**. The idea is simple: instead of splitting the input object into a finite set of pieces, we assume an underlying continuous domain: *e.g.*, text may be represented as a function $V : S \to \mathbb{R}^D$ that maps points in the real line ($S \subseteq \mathbb{R}$, continuous time) onto a $D$-dimensional vector representation, representing the "semantics" of the text evolving over time; images may be regarded as a smooth function in 2D ($S \subseteq \mathbb{R}^2$), instead of being split into regions in a grid.

Instead of scores $[f_1, \ldots, f_L]$, we now have a **score function** $f : S \to \mathbb{R}$, which we map to a probability density $p \in \mathcal{M}_+^1(S)$. This density is used in tandem with the value mapping $V : S \to \mathbb{R}^D$ to obtain a context vector $c = \mathbb{E}_p[V(t)] \in \mathbb{R}^D$. Since $\mathcal{M}_+^1(S)$ may be infinite dimensional, we need to parametrize $f$, $p$, and $V$ to be able to compute in a finite-dimensional parametric space.

**Building attention mechanisms.** We represent $f$ and $V$ using basis functions, $\phi : S \to \mathbb{R}^M$ and $\psi : S \to \mathbb{R}^N$, defining $f_\theta(t) = \theta^\top \phi(t)$ and $V_B(t) = B\psi(t)$, where $\theta \in \mathbb{R}^M$ and $B \in \mathbb{R}^{D \times N}$. The score function $f_\theta$ is mapped into a probability density $p := \hat{p}_\Omega[f_\theta]$, from which we compute the context vector as $c = \mathbb{E}_p[V_B(t)] = Br$, with $r = \mathbb{E}_p[\psi(t)]$. Summing up yields the following:

> **Definition 2.** *Let $\langle S, \Omega, \phi, \psi \rangle$ be a tuple with $\Omega : \mathcal{M}_+^1(S) \to \mathbb{R}$, $\phi : S \to \mathbb{R}^M$, and $\psi : S \to \mathbb{R}^N$. An **attention mechanism** is a mapping $\rho : \Theta \subseteq \mathbb{R}^M \to \mathbb{R}^N$, defined as:*
> $$\rho(\theta) = \mathbb{E}_p[\psi(t)], \tag{15}$$
> *with $p = \hat{p}_\Omega[f_\theta]$ and $f_\theta(t) = \theta^\top \phi(t)$. If $\Omega = \Omega_\alpha$, we call this **entmax** attention, denoted as $\rho_\alpha$. The values $\alpha = 1$ and $\alpha = 2$ lead to softmax and sparsemax attention, respectively.*

Note that, if $S = \{1, ..., L\}$ and $\phi(k) = \psi(k) = e_k$ (Euclidean canonical basis), we recover the discrete attention of Bahdanau et al. [14]. Still in the finite case, if $\phi(k)$ and $\psi(k)$ are key and value vectors and $\theta$ is a query vector, this recovers the key-value attention of Vaswani et al. [28].

On the other hand, for $S = \mathbb{R}^D$ and $\phi(t) = [t, \mathrm{vec}(tt^\top)]$, we obtain new attention mechanisms (assessed experimentally for the 1D and 2D cases in §4): for $\alpha = 1$, the underlying density $p$ is Gaussian, and for $\alpha = 2$, it is a truncated paraboloid (see §2.4). In both cases, we show (App. G) that the expectation (15) is tractable (1D) or simple to approximate numerically (2D) if $\psi$ are Gaussian RBFs, and we use this fact in §4 (see Alg. 1 for pseudo-code for the case $\alpha = 1$).

**Defining the value function $V_B(t)$.** In many problems, the input is a discrete sequence of observations (*e.g.*, text) or it was discretized (*e.g.*, images), at locations $\{t_\ell\}_{\ell=1}^L$. To turn it into a continuous signal, we need to smooth and interpolate these observations. If we start with a discrete encoder representing the input as a matrix $H \in \mathbb{R}^{D \times L}$, one way of obtaining a value mapping $V_B : S \to \mathbb{R}^D$ is by "approximating" $H$ with *multivariate ridge regression*. With $V_B(t) = B\psi(t)$, and packing the basis vectors $\psi(t_\ell)$ as columns of matrix $F \in \mathbb{R}^{N \times L}$, we obtain:

$$B^\star = \arg\min_B \|BF - H\|_F^2 + \lambda \|B\|_F^2 = HF^\top(FF^\top + \lambda I_N)^{-1} = HG, \tag{16}$$

where $\|.\|_F$ is the Frobenius norm, and the $L \times N$ matrix $G = F^\top (FF^\top + \lambda I_N)^{-1}$ depends only on the values of the basis functions at discrete time steps and can be obtained off-line for different input lenghts $L$. The result is an expression for $V_B$ with $ND$ coefficients, cheaper than $H$ if $N \ll L$.

## 3.2 Gradient backpropagation with continuous attention

The next proposition, based on Proposition 2 and proved in App. F, allows backpropagating over continuous entmax attention mechanisms. We define, for $\beta \geq 0$, a *generalized $\beta$-covariance*,

$$\mathrm{cov}_{p,\beta}[\phi(t), \psi(t)] \ = \ \|p\|_\beta^\beta \times \left( \mathbb{E}_{\tilde{p}_\beta}\left[\phi(t)\psi(t)^\top\right] - \mathbb{E}_{\tilde{p}_\beta}[\phi(t)]\,\mathbb{E}_{\tilde{p}_\beta}[\psi(t)]^\top \right), \tag{17}$$

where $\tilde{p}_\beta$ is the $\beta$-escort distribution in (5). For $\beta = 1$, we have the usual covariance; for $\beta = 0$, we get a covariance taken w.r.t. a uniform density on the support of $p$, scaled by $|\mathrm{supp}(p)|$.

**Proposition 3.** *Let $p = \hat{p}_{\Omega_\alpha}[f_\theta]$ with $f_\theta(t) = \theta^\top \phi(t)$. The Jacobian of the $\alpha$-entmax is:*

$$J_{\rho_\alpha}(\theta) = \frac{\partial \rho_\alpha(\theta)}{\partial \theta} = \mathrm{cov}_{p, 2-\alpha}(\phi(t), \psi(t)). \tag{18}$$

Note that in the finite case, (18) reduces to the expressions in (14) for softmax and sparsemax.

**Example: Gaussian RBFs.** As before, let $S = \mathbb{R}^D$, $\phi(t) = [t, \mathrm{vec}(tt^\top)]$, and $\psi_j(t) = \mathcal{N}(t; \mu_j, \Sigma_j)$. For $\alpha = 1$, we obtain closed-form expressions for the expectation (15) and the Jacobian (18), for any $D \in \mathbb{N}$: $\hat{p}_\Omega[f_\theta]$ is a Gaussian, the expectation (15) is the integral of a product of Gaussians, and the covariance (18) involves first- and second-order Gaussian moments. Pseudo-code for the case $\alpha = 1$ is shown as Alg. 1. For $\alpha = 2$, $\hat{p}_\Omega[f_\theta]$ is a truncated paraboloid. In the 1D case, both (15) and (18) can be expressed in closed form in terms of the erf function. In the 2D case, we can reduce the problem to 1D integration using the change of variable formula and working with polar coordinates. See App. G for details.

We use the facts above in the experimental section (§4), where we experiment with continuous variants of softmax and sparsemax attentions in natural language processing and vision applications.

## 4 Experiments

As proof of concept, we test our continuous attention mechanisms on three tasks: document classification, machine translation, and visual question answering (more experimental details in App. H).

**Document classification.** Although textual data is fundamentally discrete, modeling long documents as a continuous signal may be advantageous, due to smoothness and independence of length. To test this hypothesis, we use the IMDB movie review dataset [29], whose inputs are documents (280 words on average) and outputs are sentiment labels (positive/negative). Our baseline is a biLSTM with discrete attention. For our continuous attention models, we normalize the document length $L$ into the unit interval $[0, 1]$, and use $f(t) = -(t-\mu)^2/2\sigma^2$ as the score function, leading to a 1D Gaussian ($\alpha = 1$) or truncated parabola ($\alpha = 2$) as the attention density. We compare three attention variants: **discrete attention** with softmax [14] and sparsemax [6]; **continuous attention**, where a CNN and max-pooling yield a document representation $v$ from which we compute $\mu = \mathrm{sigmoid}(w_1^\top v)$ and $\sigma^2 = \mathrm{softplus}(w_2^\top v)$; and **combined attention**, which obtains $p \in \triangle^L$ from discrete attention, computes $\mu = \mathbb{E}_p[\ell/L]$ and $\sigma^2 = \mathbb{E}_p[(\ell/L)^2] - \mu^2$, applies the continuous attention, and sums the two context vectors (this model has the same number of parameters as the discrete attention baseline).

Table 1 shows accuracies for different numbers $N$ of Gaussian RBFs. The accuracies of the individual models are similar, suggesting that continuous attention is as effective as its discrete counterpart, despite having fewer basis functions than words, *i.e.*, $N \ll L$. Among the continuous variants, the sparsemax outperforms the softmax, except for $N = 64$. We also see that a large $N$ is not necessary to obtain good results, which is encouraging for tasks with long sequences. Finally, combining discrete and continuous sparsemax produced the best results, without increasing the number of parameters.

**Machine translation.** We use the De→En IWSLT 2017 dataset [30], and a biLSTM model with discrete softmax attention as a baseline. For the continuous attention models, we use the combined

Table 1: Results on IMDB in terms of accuracy (%). For the continuous attentions, we used $N \in \{32, 64, 128\}$ Gaussian RBFs $\mathcal{N}(t, \tilde{\mu}, \tilde{\sigma}^2)$, with $\tilde{\mu}$ linearly spaced in $[0, 1]$ and $\tilde{\sigma} \in \{.1, .5\}$.

| ATTENTION | $\bar{L} \approx 280$ |
| --- | --- |
| Discrete softmax | 90.78 |
| Discrete sparsemax | 90.58 |

| ATTENTION | $N = 32$ | $N = 64$ | $N = 128$ |
| --- | --- | --- | --- |
| Continuous softmax | 90.20 | 90.68 | 90.52 |
| Continuous sparsemax | 90.52 | 89.63 | 90.90 |
| Disc. + Cont. softmax | 90.98 | 90.69 | 89.62 |
| Disc. + Cont. sparsemax | **91.10** | **91.18** | **90.98** |

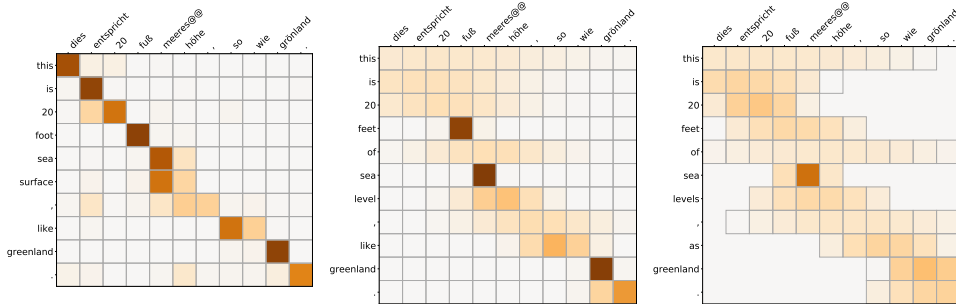

Figure 2: Attention maps in machine translation: discrete (left), continuous softmax (middle), and continuous sparsemax (right), for a sentence in the De-En IWSLT17 validation set. In the rightmost plot, the selected words are the ones with positive density. In the test set, these models attained BLEU scores of 23.92 (discrete), 24.00 (continuous softmax), and 24.25 (continuous sparsemax).

attention setting described above, with 30 Gaussian RBFs and $\tilde{\mu}$ linearly spaced in $[0, 1]$ and $\tilde{\sigma} \in \{.03, .1, .3\}$. The results (caption of Fig. 2) show a slight benefit in the combined attention over discrete attention only, without any additional parameters. Fig. 2 shows heatmaps for the different attention mechanisms on a De→En sentence. The continuous mechanism tends to have attention means close to the diagonal, adjusting the variances based on alignment confidence or when a larger context is needed (*e.g.*, a peaked density for the target word "sea", and a flat one for "of").

**Visual QA.** Finally, we report experiments with 2D continuous attention on visual question answering, using the VQA-v2 dataset [31] and a modular co-attention network as a baseline [32].[4] The discrete attention model attends over a 14×14 grid.[5] For continuous attention, we normalize the image size into the unit square $[0, 1]^2$. We fit a 2D Gaussian ($\alpha = 1$) or truncated paraboloid ($\alpha = 2$) as the attention density; both correspond to $f(t) = -\frac{1}{2}(t - \mu)^\top \Sigma^{-1}(t - \mu)$, with $\Sigma \succ 0$. We use the mean and variance according to the discrete attention probabilities and obtain $\mu$ and $\Sigma$ with moment matching. We use $N = 100 \ll 14^2$ Gaussian RBFs, with $\tilde{\mu}$ linearly spaced in $[0, 1]^2$ and $\tilde{\Sigma} = 0.001 \cdot I$. Overall, the number of neural network parameters is the same as in discrete attention.

The results in Table 2 show similar accuracies for all attention models, with a slight advantage for continuous softmax. Figure 3 shows an example (see App. H for more examples and some failure cases): in the baseline model, the discrete attention is too scattered, possibly mistaking the lamp with a TV screen. The continuous attention models focus on the right region and answer the question correctly, with continuous sparsemax enclosing all the relevant information in its supporting ellipse.

## 5 Related Work

**Relation to the Tsallis maxent principle.** Our paper unifies two lines of work: deformed exponential families from statistical physics [13, 10, 25], and sparse alternatives to softmax recently proposed in the machine learning literature [6, 8, 7], herein extended to continuous domains. This link may be fruitful for future research in both fields. While most prior work is focused on heavy-tailed

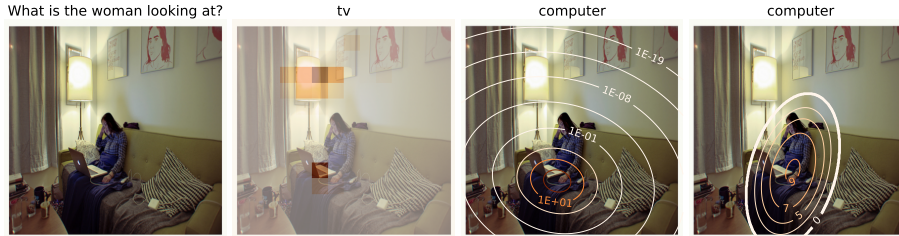

Figure 3: Attention maps for an example in VQA-v2: original image, discrete attention, continuous softmax, and continuous sparsemax. The latter encloses all probability mass within the outer ellipse.

Table 2: Accuracies of different models on the *test-dev* and *test-standard* splits of VQA-v2.

| ATTENTION | Test-Dev | | | | Test-Standard | | | |
|---|---|---|---|---|---|---|---|---|
| | Yes/No | Number | Other | Overall | Yes/No | Number | Other | Overall |
| Discrete softmax | 83.40 | 43.59 | 55.91 | 65.83 | 83.47 | 42.99 | 56.33 | 66.13 |
| 2D continuous softmax | 83.40 | 44.80 | 55.88 | **65.96** | 83.79 | 44.33 | 56.04 | **66.27** |
| 2D continuous sparsemax | 83.10 | 44.12 | 55.95 | 65.79 | 83.38 | 43.91 | 56.14 | 66.10 |

distributions ($\alpha < 1$), we focus instead on light-tailed, sparse distributions, the other side of the spectrum ($\alpha > 1$). See App. C for the relation to the Tsallis maxent principle.

**Continuity in other architectures and dimensions.** In our paper, we consider attention networks exhibiting temporal/spatial continuity in the input data, be it text (1D) or images (2D). Recent work propose continuous-domain CNNs for 3D structures like point clouds and molecules [34, 35]. The dynamics of continuous-time RNNs have been studied in [36], and similar ideas have been applied to irregularly sampled time series [37]. Other recently proposed frameworks produce continuous variants in other dimensions, such as network depth [38], or in the target domain for machine translation tasks [39]. Our continuous attention networks can be used in tandem with these frameworks.

**Gaussian attention probabilities.** Cordonnier et al. [40] analyze the relationship between (discrete) attention and convolutional layers, and consider spherical Gaussian attention probabilities as relative positional encodings. By contrast, our approach removes the need for positional encodings: by converting the input to a function on a predefined continuous space, positions are encoded *implicitly*, not requiring explicit positional encoding. Gaussian attention has also been hard-coded as input-agnostic self-attention layers in transformers for machine translation tasks by You et al. [41]. Finally, in their DRAW architecture for image generation, Gregor et al. [42, §3.1] propose a selective attention component which is parametrized by a spherical Gaussian distribution.

## 6 Conclusions and Future Work

We proposed extensions to regularized prediction maps, originally defined on finite domains, to arbitrary measure spaces (§2). With Tsallis $\alpha$-entropies for $\alpha > 1$, we obtain sparse families, whose members can have zero tails, such as triangular or truncated parabola distributions. We then used these distributions to construct continuous attention mechanisms (§3). We derived their Jacobians in terms of generalized covariances (Proposition 3), allowing for efficient forward and backward propagation. Experiments for 1D and 2D cases were shown on attention-based text classification, machine translation, and visual question answering (§4), with encouraging results.

There are many avenues for future work. As a first step, we considered unimodal distributions only (Gaussian, truncated paraboloid), for which we show that the forward and backpropagation steps have closed form or can be reduced to 1D integration. However, there are applications in which multiple attention modes are desirable. This can be done by considering mixtures of distributions, multiple attention heads, or sequential attention steps. Another direction concerns combining our continuous attention models with other spatial/temporal continuous architectures for CNNs and RNNs [34–36] or with continuity in other dimensions, such as depth [38] or output space [39].

## Broader Impact

We discuss the broader impact of our work, including ethical aspects and future societal consequences. Given the early stage of our work and its predominantly theoretical nature, the discussion is mostly speculative.

The continuous attention models developed in our work can be used in a very wide range of applications, including natural language processing, computer vision, and others. For many of these applications, current state-of-the-art models use discrete softmax attention, whose interpretation capabilities have been questioned in prior work [43–45]. Our models can potentially lead to more interpretable decisions, since they lead to less scattered attention maps (as shown in our Figures 2–3) and are able to select contiguous text segments or image regions. As such, they may provide better inductive bias for interpretation.

In addition, our attention densities using Gaussian and truncated paraboloids include a variance term, being potentially useful as a measure of confidence—for example, a large ellipse in an image may indicate that the model had little confidence about where it should attend to answer a question, while a small ellipse may denote high confidence on a particular object.

We also see opportunities for research connecting our work with other continuous models [34, 35, 38] leading to end-to-end continuous models which, by avoiding discretization, have the potential to be less susceptible to adversarial attacks via input perturbations. Outside the machine learning field, the links drawn in §2 between sparse alternatives to softmax and models used in non-extensive (Tsallis) statistical physics suggest a potential benefit in that field too.

Note, however, that our work is a first step into all these directions, and as such further investigation will be needed to better understand the potential benefits. We strongly recommend carrying out user studies before deploying any such system, to better understand the benefits and risks. Some of the examples in App. H may help understand potential failure modes.

We should also take into account that, for any computer vision model, there are important societal risks related to privacy-violating surveillance applications. Continuous attention holds the promise to scale to larger and multi-resolution images, which may, in the longer term, be deployed in such undesirable domains. Ethical concerns hold for natural language applications such as machine translation, where biases present in data can be arbitrarily augmented or hidden by machine learning systems. For example, our natural language processing experiments mostly use English datasets (as a target language in machine translation, and in document classification). Further work is needed to understand if our findings generalize to other languages. Likewise, in the vision experiments, the VQA-v2 dataset uses COCO images, which have documented biases [46]. In line with the fundamental scope and early stage of this line of research, we deliberately choose applications on standard benchmark datasets, in an attempt to put as much distance as possible from malevolent applications. Finally, although we chose the most widely used evaluation metrics for each task (accuracy for document classification and visual question answering, BLEU for machine translation), these metrics do not always capture performance quality—for example, BLEU in machine translation is far from being a perfect metric.

The data, memory, and computation requirements for training systems with continuous attention do not seem considerably higher than the ones which use discrete attention. On the other hand, for NLP applications, our approach has the potential to better compress sequential data, by using fewer basis functions than the sequence length (as suggested by our document classification experiments). While there is nothing specific about our research that poses environmental concerns or that promises to alleviate such concerns, our models share the same problematic property as other neural network models in terms of their energy consumption to train models and tune hyperparameters [47].

## Acknowledgments and Disclosure of Funding

This work was supported by the European Research Council (ERC StG DeepSPIN 758969), by the P2020 program MAIA (contract 045909), and by the Fundação para a Ciência e Tecnologia through contract UIDB/50008/2020. We would like to thank Pedro Martins, Zita Marinho, and the anonymous reviewers for their helpful feedback.

## Footnotes

[1]This entropy is normally defined up to a constant, often presented without the $\frac{1}{\alpha}$ factor. We use the same definition as Blondel et al. [7, §4.3] for convenience.

[2]This should not be confused with sparsity-inducing distributions [22, 23].

[3]Unfortunately, the literature is inconsistent in defining these coefficients. Our $\alpha$ matches that of Blondel et al. [7]; Tsallis' $q$ equals $2 - \alpha$; this family is also related to Amari's $\alpha$-divergences, but their $\alpha = 2q - 1$. Inconsistent definitions have also been proposed for $q$-exponential families regarding how they are normalized; for example, the Tsallis maxent principle leads to a different definition. See App. C for a detailed discussion.

[4]Software code is available at https://github.com/deep-spin/mcan-vqa-continuous-attention.

[5]An alternative would be bounding box features from an external object detector [33]. We opted for grid regions to check if continuous attention has the ability to detect relevant objects on its own. However, our method can handle bounding boxes too, if the $\{t_\ell\}_{\ell=1}^L$ coordinates in the regression (16) are placed on those regions.

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
