[Supplementary Material]

# Supplemental Material

## A  Differential Negentropy and Boltzmann-Gibbs distributions

We adapt a proof from Cover and Thomas [17]. Let $\Omega$ be the Shannon negentropy, which is proper, lower semi-continuous, and strictly convex [48, example 9.41], and let

$$\text{KL}(p\|q) := \int_S p(t) \log \frac{p(t)}{q(t)}$$

be the Kullback-Leibler divergence between distributions $p$ and $q$ (which is always non-negative and equals 0 iff $p = q$). Take $q(t) = \frac{\exp(f(t))}{\int_S \exp(f(t'))d\nu(t')} = \exp(f(t) - A(f))$ as in (2), where $A(f)$ is the log-partition function.

We have, for any $p \in \mathcal{M}_+^1(S)$:

$$
\begin{aligned}
0 &\leq \text{KL}(p\|q) = \int_S p(t) \log \frac{p(t)}{q(t)} = \Omega(p) - \int_S p(t) \log q(t) = \Omega(p) - \int_S p(t)(f(t) - A(f)) \\
&= \Omega(p) - \mathbb{E}_p[f(t)] + A(f).
\end{aligned}
\tag{19}
$$

Therefore, we have, for any $p \in \mathcal{M}_+^1(S)$, that

$$\mathbb{E}_p[f(t)] - \Omega(p) \leq A(f), \tag{20}$$

with equality if and only if $p = q$. Since the right hand side is constant with respect to $p$, we have that the posited $q$ must be the maximizer of (1).

## B  Tsallis Negentropy and Sparse Distributions

### B.1  Shannon as a limit case of Tsallis when $\alpha \to 1$

We show that $\lim_{\alpha \to 1} \Omega_\alpha(p) = \Omega_1(p)$ for any $p \in \mathcal{M}_+^1(S)$. From (6), it suffices to show that $\lim_{\beta \to 1} \log_\beta(u) = \log(u)$ for any $u \geq 0$. Let $g(\beta) := u^{1-\beta} - 1$, and $h(\beta) := 1 - \beta$. Observe that

$$\lim_{\beta \to 1} \log_\beta(u) = \lim_{\beta \to 1} \frac{g(\beta)}{h(\beta)} = \frac{g(1)}{h(1)} = \frac{0}{0},$$

so we are in an indeterminate case. We take the derivatives of $g$ and $h$:

$$g'(\beta) = \left( \exp(\log u^{1-\beta}) \right)' = \exp(\log u^{1-\beta}) \cdot ((1-\beta)\log u)' = -u^{1-\beta} \log u, \tag{21}$$

and $h'(\beta) = -1$. From l'Hôpital's rule,

$$\lim_{\beta \to 1} \frac{g(\beta)}{h(\beta)} = \lim_{\beta \to 1} \frac{g'(\beta)}{h'(\beta)} = \log u. \tag{22}$$

### B.2  Proof of Proposition 1

The proof of Proposition 1 is similar to the one in §A, replacing the KL divergence by the Bregman divergence induced by $\Omega_\alpha$, and using an additional bound. Let

$$B_{\Omega_\alpha}(p, q) := \Omega_\alpha(p) - \Omega_\alpha(q) - \langle \nabla \Omega_\alpha(q), p - q \rangle$$

be the (functional) Bregman divergence between distributions $p$ and $q$ induced by $\Omega_\alpha$, and let

$$q(t) = \exp_{2-\alpha}(f(t) - A_\alpha(f)) = [1 + (\alpha - 1)(f(t) - A_\alpha(f))]_+^{\frac{1}{\alpha-1}}.$$

Note that, from (6),

$$\left( \nabla_q \Omega_\alpha(q) \right)(t) = \frac{q(t)^{\alpha-1}}{\alpha - 1}.$$

From the non-negativity of the Bregman divergence [49], we have, for any $p \in M_+^1(S)$:

$$
\begin{aligned}
0 \quad \leq^{(a)} \quad & B_{\Omega_\alpha}(p,q) \\
= \quad & \Omega_\alpha(p) - \Omega_\alpha(q) - \langle \nabla\Omega_\alpha(q), p - q \rangle \\
= \quad & \Omega_\alpha(p) - \Omega_\alpha(q) - \int_S \frac{q(t)^{\alpha-1}}{\alpha-1}(p(t) - q(t)) \\
= \quad & \Omega_\alpha(p) - \Omega_\alpha(q) - \underbrace{\mathbb{E}_p[[f(t) - A_\alpha(f) + (\alpha-1)^{-1}]_+]}_{\geq \mathbb{E}_p[f(t) - A_\alpha(f) + (\alpha-1)^{-1}]} + \frac{1}{\alpha-1}\int_S q(t)^\alpha \\
\leq^{(b)} \quad & \Omega_\alpha(p) - \Omega_\alpha(q) - \mathbb{E}_p[f(t) - A_\alpha(f) + (\alpha-1)^{-1}] + \frac{1}{\alpha-1}\int_S q(t)^\alpha \\
= \quad & \Omega_\alpha(p) - \mathbb{E}_p[f(t)] - \Omega_\alpha(q) + \underbrace{\frac{1}{\alpha-1}\left(\int_S q(t)^\alpha - 1\right)}_{=\alpha\Omega_\alpha(q)} + A_\alpha(f) \\
= \quad & \Omega_\alpha(p) - \mathbb{E}_p[f(t)] + (\alpha-1)\Omega_\alpha(q) + A_\alpha(f). \qquad (23)
\end{aligned}
$$

Therefore, we have, for any $p \in M_+^1(S)$,

$$
\mathbb{E}_p[f(t)] - \Omega_\alpha(p) \leq (\alpha-1)\Omega_\alpha(q) + A_\alpha(f), \qquad (24)
$$

with equality iff $p = q$, which leads to zero Bregman divergence (*i.e.*, a tight inequality $(a)$) and to $\mathbb{E}_p[[f(t) - A_\alpha(f) + (\alpha-1)^{-1}]_+] = \mathbb{E}_p[f(t) - A_\alpha(f) + (\alpha-1)^{-1}]$ (*i.e.*, a tight inequality $(b)$).

We can use the equality above to obtain an expression for the Fenchel conjugate $\Omega_\alpha^*(f) = \mathbb{E}_q[f(t)] - \Omega_\alpha(q)$ (*i.e.*, the value of the maximum in (1) and the right hand side in (24)):

$$
\Omega_\alpha^*(f) = (\alpha-1)\Omega_\alpha(q) + A_\alpha(f). \qquad (25)
$$

### B.3  Normalizing function $A_\alpha(f)$

Let $p = \hat{p}_{\Omega_\alpha}[f]$. The expression for $A_\alpha$ in Prop. 1 is obtained by inverting (7), yielding $A_\alpha(f) = f(t) - \log_{2-\alpha}(p(t))$, and integrating with respect to $p(t)^{2-\alpha}d\nu(t)$, leading to:

$$
\begin{aligned}
\int_S p_\theta(t)^{2-\alpha} A_\alpha(f) \quad = \quad & \int_S p(t)^{2-\alpha} f(t) - \int_S p(t)^{2-\alpha} \log_{2-\alpha}(p(t)) \\
= \quad & \int_S p_\theta(t)^{2-\alpha} f(t) - \frac{\int_S (p(t) - p(t)^{2-\alpha})}{\alpha-1} \\
= \quad & \int_S p(t)^{2-\alpha} f(t) - \frac{1}{\alpha-1} + \frac{\int_S p(t)^{2-\alpha}}{\alpha-1}, \qquad (26)
\end{aligned}
$$

from which the desired expression follows.

## C  Relation to the Tsallis Maxent Principle

We discuss here the relation between the $(2-\alpha)$-exponential family of distributions as presented in Prop. 1 and the distributions arising from the Tsallis maxent principle [13]. We put in perspective the related work in statistical physics [50, 10], information geometry [25, 18], and the discrete case presented in the machine learning literature [7, 8].

We start by noting that our $\alpha$ parameter matches the $\alpha$ used in prior machine learning literature related to the "$\alpha$-entmax transformation" [7, 8]. In the definition of Tsallis entropies (6), our $\alpha$ corresponds to the entropic index $q$ defined by Tsallis [13]. However, our $(2-\alpha)$-exponential families correspond to the $q$-exponential families as defined by Naudts [10], and to the $t$-exponential families described by Ding and Vishwanathan [12] (which include the $t$-Student distribution). The family of Amari's $\alpha$-divergences relates to this $q$ as $\alpha = 2q - 1$ [18, §4.3].

These differences in notation have historical reasons, and they are explained by the different ways in which Tsallis entropies relate to $q$-exponential families. In fact, the physics literature has defined $q$-exponential distributions in two distinct ways, as we next describe.

Note first that the $\Omega$-RPM in our Def. 1 is a generalization of the free energy variational principle, if we see $-f_\theta(t) = -\theta^\top \phi(t)$ as an energy function and $\Omega$ the entropy scaled by a temperature. Let $\Omega = \Omega_\alpha$ be the Tsallis $\alpha$-entropy. An equivalent constrained version of this problem is the maximum entropy (*maxent*) principle [51]:

$$\max_{p \in \mathcal{M}_+^1(S)} -\Omega_\alpha(p), \quad \text{s.t.} \quad \mathbb{E}_p[\phi(t)] = b. \tag{27}$$

The solution of this problem corresponds to a distribution in the $(2-\alpha)$-exponential family (7):

$$p^\star(t) = \exp_{2-\alpha}(\theta^\top \phi(t) - A_\alpha(\theta)), \tag{28}$$

for some Lagrange multiplier $\theta$.

However, this construction differs from the one by Tsallis [13] and others, who use *escort distributions* (Eq. 5) in the expectation constraints. Namely, instead of (27), they consider the problem:

$$\max_{p \in \mathcal{M}_+^1(S)} -\Omega_\alpha(p), \quad \text{s.t.} \quad \mathbb{E}_{\tilde{p}^\alpha}[\phi(t)] = b. \tag{29}$$

The solution of (29) is of the form

$$p^\star(t) = B_\alpha(\theta) \exp_\alpha(\theta^\top(\phi(t) - b)), \tag{30}$$

where $\theta$ is again a Lagrange multiplier. This is derived, for example, in [50, Eq. 15]. There are two main differences between (28) and (30):

- While (28) involves the $(2-\alpha)$-exponential, (30) involves the $\alpha$-exponential.
- In (28), the normalizing term $A_\alpha(\theta)$ is *inside* the $(2-\alpha)$-exponential. In (30), there is an normalizing factor $B_\alpha(\theta)$ *outside* the $\alpha$-exponential.

Naturally, when $\alpha = 1$, these two problems become equivalent, since an additive term inside the exponential is equivalent to a multiplicative term outside. However, this does *not* happen with $\beta$-exponentials ($\exp_\beta(u+v) \neq \exp_\beta(u)\exp_\beta(v)$ in general, for $\beta \neq 1$), and therefore these two alternative paths lead to two different definitions of $q$-exponential families. Unfortunately, both have been considered in the physics literature, under the same name, and this has been subject of debate. Quoting Naudts [10, §1]:

> *"An important question is then whether in the modification the normalization should stand in front of the deformed exponential function, or whether it should be included as $\ln Z(\beta)$ inside. From the general formalism mentioned above it follows that the latter is the right way to go."*

Throughout our paper, we use the definition of [10, 25], equivalent to the maxent problem (27).

## D  Proof of Proposition 2

We adapt the proof from Amari and Ohara [25, Theorem 5]. Note first that, for $t \in \text{supp}(p_\theta)$,

$$\begin{aligned}
\nabla_\theta p_\theta(t) &= \nabla_\theta[(\alpha-1)(\theta^\top \phi(t) - A_\alpha(\theta)) + 1]^{1/(\alpha-1)} \\
&= [(\alpha-1)(\theta^\top \phi(t) - A_\alpha(\theta)) + 1]^{(2-\alpha)/(\alpha-1)}(\phi(t) - \nabla_\theta A_\alpha(\theta)) \\
&= p_\theta(t)^{2-\alpha}(\phi(t) - \nabla_\theta A_\alpha(\theta)),
\end{aligned} \tag{31}$$

and

$$\begin{aligned}
\nabla_\theta^2 p_\theta(t) &= \nabla_\theta p_\theta^{2-\alpha}(t)(\phi(t) - \nabla_\theta A_\alpha(\theta))^\top - p_\theta^{2-\alpha}(t)\nabla_\theta^2 A_\alpha(\theta) \\
&= (2-\alpha)p_\theta^{1-\alpha}(t)\nabla_\theta p_\theta(t)(\phi(t) - \nabla_\theta A_\alpha(\theta))^\top - p_\theta^{2-\alpha}(t)\nabla_\theta^2 A_\alpha(\theta) \\
&= (2-\alpha)p_\theta(t)^{3-2\alpha}(\phi(t) - \nabla_\theta A_\alpha(\theta))(\phi(t) - \nabla_\theta A_\alpha(\theta))^\top \\
&\quad - p_\theta(t)^{2-\alpha}\nabla_\theta^2 A_\alpha(\theta).
\end{aligned} \tag{32}$$

Therefore we have:

$$0 = \nabla_\theta \underbrace{\int_S p_\theta(t)}_{=1} = \int_S \nabla_\theta p_\theta(t) = \int_S p_\theta(t)^{2-\alpha}(\phi(t) - \nabla_\theta A_\alpha(\theta)), \tag{33}$$

from which we obtain

$$\nabla_\theta A_\alpha(\theta) = \frac{\int_S p_\theta(t)^{2-\alpha}\phi(t)}{\int_S p_\theta(t)^{2-\alpha}}. \tag{34}$$

To prove that $A_\alpha(\theta)$ is convex, we will show that its Hessian is positive semidefinite. Note that

$$
\begin{aligned}
0 &= \nabla_\theta^2 \underbrace{\int_S p_\theta(t)}_{=1} = \int_S \nabla_\theta^2 p_\theta(t) \\
&= \int_S (2-\alpha)p_\theta(t)^{3-2\alpha}\big(\phi(t) - \nabla_\theta A_\alpha(\theta)\big)\big(\phi(t) - \nabla_\theta A_\alpha(\theta)\big)^\top - p_\theta(t)^{2-\alpha}\nabla_\theta^2 A_\alpha(\theta) \\
&= (2-\alpha)\int_S p_\theta(t)^{3-2\alpha}\big(\phi(t) - \nabla_\theta A_\alpha(\theta)\big)\big(\phi(t) - \nabla_\theta A_\alpha(\theta)\big)^\top \\
&\quad - \nabla_\theta^2 A_\alpha(\theta)\int_S p_\theta(t)^{2-\alpha},
\end{aligned} \tag{35}
$$

hence, for $\alpha \le 2$,

$$\nabla_\theta^2 A_\alpha(\theta) = \frac{(2-\alpha)\int_S p_\theta(t)^{3-2\alpha}\overbrace{\big(\phi(t) - \nabla_\theta A_\alpha(\theta)\big)\big(\phi(t) - \nabla_\theta A_\alpha(\theta)\big)^\top}^{\succeq 0}}{\int_S p_\theta(t)^{2-\alpha}} \succeq 0, \tag{36}$$

where we used the fact that $p_\theta(t) \ge 0$ for $t \in S$ and that integrals of positive semidefinite functions and positive semidefinite.

# E  Normalization Constants for Continuous Sparsemax Distributions

## E.1  Truncated parabola

Let $p(t) = \left[-\lambda - \frac{(t-\mu)^2}{2\sigma^2}\right]_+$ as in (10). Let us determine the constant $\lambda$ that ensures this distribution normalizes to 1. Note that $\lambda$ does not depend on the location parameter $\mu$, hence we can assume $\mu = 0$ without loss of generality. We must have $\lambda = -\frac{a^2}{2\sigma^2}$ and $1 = \int_{-a}^{a}\left(-\lambda - \frac{x^2}{2\sigma^2}\right) = -2\lambda a - \frac{a^3}{3\sigma^2} = \frac{2a^3}{3\sigma^2}$, hence $a = \left(\frac{3}{2}\sigma^2\right)^{1/3}$, which finally gives:

$$\lambda = -\frac{1}{2}\left(\frac{3}{2\sigma}\right)^{2/3}. \tag{37}$$

## E.2  Multivariate truncated paraboloid

Let $p(t) = \left[-\lambda - \frac{1}{2}(t-\mu)\Sigma^{-1}(t-\mu)\right]_+$ as in (11). Let us determine the constant $\lambda$ that ensures this distribution normalizes to 1, where we assume again $\mu = 0$ without loss of generality. To obtain $\lambda$, we start by invoking the formula for computing the volume of an ellipsoid defined by the equation $x^\top\Sigma^{-1}x \le 1$:

$$V_{\text{ell}}(\Sigma) = \frac{\pi^{n/2}}{\Gamma(n/2+1)}\det(\Sigma)^{1/2}, \tag{38}$$

where $\Gamma(t)$ is the Gamma function. Since each slice of a paraboloid is an ellipsoid, we can apply Cavalieri's principle to obtain the volume of a paraboloid $y = \frac{1}{2}x^\top\Sigma^{-1}x$ of height $h = -\lambda$ as follows:

$$
\begin{aligned}
V_{\text{par}}(h) &= \int_0^h V_{\text{ell}}(2\Sigma y)dy = \frac{(2\pi)^{n/2}\det(\Sigma)^{1/2}}{\Gamma(\frac{n}{2}+1)}\int_0^h y^{\frac{n}{2}}dy \\
&= \frac{(2\pi)^{n/2}\det(\Sigma)^{1/2}}{(\frac{n}{2}+1)\Gamma(\frac{n}{2}+1)}h^{\frac{n}{2}+1} \\
&= \frac{\sqrt{(2\pi)^n\det(\Sigma)}}{\Gamma(\frac{n}{2}+2)}h^{\frac{n}{2}+1}.
\end{aligned} \tag{39}
$$

Equating the volume to 1, we obtain $\lambda = -h$ as:

$$\lambda = -\left( \frac{\Gamma(\frac{n}{2} + 2)}{\sqrt{(2\pi)^n \det(\Sigma)}} \right)^{\frac{2}{2+n}}. \tag{40}$$

### E.3 Triangular

Let $p(t) = \left[ -\lambda - \frac{|t-\mu|}{b} \right]_+$ as in (12). Let us determine the constant $\lambda$ that ensures this distribution normalizes to 1. Assuming again $\mu = 0$ without loss of generality, we must have $\lambda = -\frac{a}{b}$ and $1 = \int_{-a}^{a} \left( -\lambda - \frac{|x|}{b} \right) = -2\lambda a - \frac{a^2}{b} = \frac{a^2}{b}$, hence $a = \sqrt{b}$, which finally gives $\lambda = -b^{-1/2}$.

### E.4 Location-scale families

We first show that $a$ is the solution of the equation $ag'(a) - g(a) + g(0) = \frac{1}{2}$. From symmetry around $\mu$, we must have

$$\frac{1}{2} = \int_{\mu}^{\mu+a\sigma} \left( \frac{1}{\sigma} g'(a) - \frac{1}{\sigma} g'\left( \frac{t-\mu}{\sigma} \right) \right) dt = \int_{0}^{a} (g'(a) - g'(s)) \, ds = ag'(a) - g(a) + g(0), \tag{41}$$

where we made a variable substitution $s = (t - \mu)/\sigma$, which proves the desired result. Now we show that a solution always exists if $g$ is strongly convex, *i.e.*, if there is some $\gamma > 0$ such that $g(0) \geq g(s) - sg'(s) + \frac{\gamma}{2}s^2$ for any $s \geq 0$. Let $F(s) := sg'(s) - g(s) + g(0)$. We want to show that the equation $F(a) = \frac{1}{2}$ has a solution. Since $g$ is continuously differentiable, $F$ is continuous. From the strong convexity of $g$, we have that $F(s) \geq \frac{\gamma}{2}s^2$ for any $s \geq 0$, which implies that $\lim_{s \to +\infty} F(s) = +\infty$. Therefore, since $F(0) = 0$, we have by the intermediate value theorem that there must be some $a$ such that $F(a) = \frac{1}{2}$.

## F    Proof of Proposition 3

We have

$$
\begin{aligned}
\nabla_\theta \mathbb{E}_p[\psi_i(t)] &= \nabla_\theta \int_S p_\theta(t)\psi_i(t) = \int_S \nabla_\theta p_\theta(t)\psi_i(t) \\
&= \int_S p_\theta^{2-\alpha}(t) \nabla_\theta \log_{2-\alpha}(p_\theta(t))\psi_i(t) \\
&= \int_S p_\theta^{2-\alpha}(t) \nabla_\theta (\theta^\top \phi(t) - A_\alpha(\theta))\psi_i(t) \\
&= \int_S p_\theta^{2-\alpha}(t)(\phi(t) - \nabla_\theta A_\alpha(\theta))\psi_i(t). \tag{42}
\end{aligned}
$$

Using the expression for $\nabla_\theta A_\alpha(\theta)$ from Proposition 2 yields the desired result.

## G    Continuous Attention with Gaussian RBFs

We derive expressions for the evaluation and gradient computation of continuous attention mechanisms where $\psi(t)$ are Gaussian radial basis functions, both for the softmax ($\alpha = 1$) and sparsemax ($\alpha = 2$) cases. For softmax, we show closed-form expressions for any number of dimensions (including the 1D and 2D cases). For sparsemax, we derive closed-form expressions for the 1D case, and we reduce the 2D case to a univariate integral on an interval, easy to compute numerically.

This makes it possible to plug both continuous attention mechanisms in neural networks and learn them end-to-end with the gradient backpropagation algorithm.

### G.1    Continuous softmax ($\alpha = 1$)

We derive expressions for continuous softmax for multivariate Gaussians in $\mathbb{R}^D$. This includes the 1D and 2D cases, where $D \in \{1, 2\}$.

If $S = \mathbb{R}^D$, for $\phi(t) = [t, tt^\top]$, the distribution $p = \hat{p}_{\Omega_1}[f_\theta]$, with $f_\theta(t) = \theta^\top \phi(t)$, is a multivariate Gaussian where the mean $\mu$ and the covariance matrix $\Sigma$ are related to the canonical parameters as $\theta = [\Sigma^{-1}\mu, -\frac{1}{2}\Sigma^{-1}]$.

We derive closed form expressions for the attention mechanism output $\rho_1(\theta) = \mathbb{E}_p[\psi(t)]$ in (15) and for its Jacobian $J_{\rho_1}(\theta) = \mathrm{cov}_{p,1}(\phi(t), \psi(t))$ in (18), when $\psi(t)$ are Gaussian RBFs, i.e., each $\psi_j$ is of the form $\psi_j(t) = \mathcal{N}(t; \mu_j, \Sigma_j)$.

**Forward pass.** Each coordinate of the attention mechanism output becomes the integral of a product of Gaussians,

$$\mathbb{E}_p[\psi_j(t)] = \int_{\mathbb{R}^D} \mathcal{N}(t; \mu, \Sigma)\mathcal{N}(t; \mu_j, \Sigma_j). \tag{43}$$

We use the fact that the product of two Gaussians is a scaled Gaussian:

$$\mathcal{N}(t; \mu, \Sigma)\mathcal{N}(t; \mu_j, \Sigma_j) = \tilde{s}\mathcal{N}(t; \tilde{\mu}, \tilde{\Sigma}), \tag{44}$$

where

$$\tilde{s} = \mathcal{N}(\mu; \mu_j, \Sigma + \Sigma_j), \qquad \tilde{\Sigma} = (\Sigma^{-1} + \Sigma_j^{-1})^{-1}, \qquad \tilde{\mu} = \tilde{\Sigma}(\Sigma^{-1}\mu + \Sigma_j^{-1}\mu_j). \tag{45}$$

Therefore, the forward pass can be computed as:

$$\begin{aligned}
\mathbb{E}_p[\psi_j(t)] &= \tilde{s} \int_{\mathbb{R}^D} \mathcal{N}(t; \tilde{\mu}, \tilde{\Sigma}) = \tilde{s} \\
&= \mathcal{N}(\mu; \mu_j, \Sigma + \Sigma_j).
\end{aligned} \tag{46}$$

**Backward pass.** To compute the backward pass, we have that each row of the Jacobian $J_{\rho_1}(\theta)$ becomes a first or second moment under the resulting Gaussian:

$$\begin{aligned}
\mathrm{cov}_{p,1}(t, \psi_j(t)) &= \mathbb{E}_p[t\psi_j(t)] - \mathbb{E}_p[t]\mathbb{E}_p[\psi_j(t)] \\
&= \int_{\mathbb{R}^D} t\mathcal{N}(t; \mu, \Sigma)\mathcal{N}(t; \mu_j, \Sigma_j) - \tilde{s}\mu \\
&= \tilde{s} \int_{\mathbb{R}^D} t\mathcal{N}(t; \tilde{\mu}, \tilde{\Sigma}) - \tilde{s}\mu \\
&= \tilde{s}(\tilde{\mu} - \mu),
\end{aligned} \tag{47}$$

and, noting that $\Sigma = \mathbb{E}[(t - \mu)(t - \mu)^\top] = \mathbb{E}[tt^\top] - \mu\mu^\top$,

$$\begin{aligned}
\mathrm{cov}_{p,1}(tt^\top, \psi_j(t)) &= \mathbb{E}_p[tt^\top \psi_j(t)] - \mathbb{E}_p[tt^\top]\mathbb{E}_p[\psi_j(t)] \\
&= \int_{\mathbb{R}^D} tt^\top \mathcal{N}(t; \mu, \Sigma)\mathcal{N}(t; \mu_j, \Sigma_j) - \tilde{s}(\Sigma + \mu\mu^\top) \\
&= \tilde{s} \int_{\mathbb{R}^D} tt^\top \mathcal{N}(t; \tilde{\mu}, \tilde{\Sigma}) - \tilde{s}(\Sigma + \mu\mu^\top) \\
&= \tilde{s}(\tilde{\Sigma} + \tilde{\mu}\tilde{\mu}^\top) - \tilde{s}(\Sigma + \mu\mu^\top) \\
&= \tilde{s}(\tilde{\Sigma} + \tilde{\mu}\tilde{\mu}^\top - \Sigma - \mu\mu^\top).
\end{aligned} \tag{48}$$

### G.2 Continuous sparsemax in 1D ($\alpha = 2$, $D = 1$)

With $\phi(t) = [t, t^2]$, the distribution $p = \hat{p}_{\Omega_2}[f_\theta]$, with $f_\theta(t) = \theta^\top \phi(t)$, becomes a truncated parabola where $\mu$ and $\sigma^2$ are related to the canonical parameters as above, i.e., $\theta = [\frac{\mu}{\sigma^2}, -\frac{1}{2\sigma^2}]$.

We derive closed form expressions for the attention mechanism output $\rho_2(\theta) = \mathbb{E}_p[\psi(t)]$ in (15) and its Jacobian $J_{\rho_2}(\theta) = \frac{\partial \rho_2(\theta)}{\partial \theta} = \mathrm{cov}_{p,2}(\phi(t), \psi(t))$ in (18) when $\psi(t)$ and Gaussian RBFs, i.e., each $\psi_j$ is of the form $\psi_j(t) = \mathcal{N}(t; \mu_j, \sigma_j^2)$.

**Forward pass.** Each coordinate of the attention mechanism output becomes:

$$
\begin{aligned}
\mathbb{E}_p[\psi_j(t)] &= \int_{\mu-a}^{\mu+a} \left( -\lambda - \frac{(t-\mu)^2}{2\sigma^2} \right) \mathcal{N}(t; \mu_j, \sigma_j^2) \\
&= \int_{\frac{\mu-\mu_j-a}{\sigma_j}}^{\frac{\mu-\mu_j+a}{\sigma_j}} \frac{1}{\sigma_j} \left( -\lambda - \frac{(\sigma_j t + \mu_j - \mu)^2}{2\sigma^2} \right) \mathcal{N}(s; 0, 1) ds,
\end{aligned}
\tag{49}
$$

where $a = (\frac{3}{2}\sigma^2)^{1/3}$ and $\lambda = -\frac{a^2}{2\sigma^2} = -\frac{1}{2}(\frac{3}{2\sigma})^{2/3}$, as stated in (37), and we made the substitution $s = \frac{t-\mu_j}{\sigma_j}$. We use the fact that, for any $u, v \in \mathbb{R}$ such that $u \le v$:

$$
\begin{aligned}
\int_u^v \mathcal{N}(t; 0, 1) &= \frac{1}{2} \left( \text{erf}\left( \frac{v}{\sqrt{2}} \right) - \text{erf}\left( \frac{u}{\sqrt{2}} \right) \right), \\
\int_u^v t\mathcal{N}(t; 0, 1) &= -\mathcal{N}(v; 0, 1) + \mathcal{N}(u; 0, 1), \\
\int_u^v t^2\mathcal{N}(t; 0, 1) &= \frac{1}{2} \left( \text{erf}\left( \frac{v}{\sqrt{2}} \right) - \text{erf}\left( \frac{u}{\sqrt{2}} \right) \right) - v\mathcal{N}(v; 0, 1) + u\mathcal{N}(u; 0, 1),
\end{aligned}
\tag{50}
$$

from which the expectation (49) can be computed directly.

**Backward pass.** Since $|\text{supp}(p)| = 2a$, we have from (17) and (50) that each row of the Jacobian $J_{\rho_2}(\theta)$ becomes:

$$
\begin{aligned}
&\text{cov}_{p,2}(t, \psi_j(t)) = \\
&\int_{\mu-a}^{\mu+a} t\mathcal{N}(t; \mu_j, \sigma_j^2) - \frac{1}{2a} \left( \int_{\mu-a}^{\mu+a} t \right) \left( \int_{\mu-a}^{\mu+a} \mathcal{N}(t; \mu_j, \sigma_j^2) \right) \\
&= \int_{\frac{\mu-\mu_j-a}{\sigma_j}}^{\frac{\mu-\mu_j+a}{\sigma_j}} (\mu_j + \sigma_j s)\mathcal{N}(s; 0, 1) - \underbrace{\frac{1}{2a} \left( \frac{(\mu+a)^2}{2} - \frac{(\mu-a)^2}{2} \right)}_{=\mu} \left( \int_{\frac{\mu-\mu_j-a}{\sigma_j}}^{\frac{\mu-\mu_j+a}{\sigma_j}} \mathcal{N}(s; 0, 1) \right) \\
&= (\mu_j - \mu) \int_{\frac{\mu-\mu_j-a}{\sigma_j}}^{\frac{\mu-\mu_j+a}{\sigma_j}} \mathcal{N}(s; 0, 1) + \sigma_j \int_{\frac{\mu-\mu_j-a}{\sigma_j}}^{\frac{\mu-\mu_j+a}{\sigma_j}} s\mathcal{N}(s; 0, 1) \\
&= \frac{\mu_j - \mu}{2} \left( \text{erf}\left( \frac{\mu - \mu_j + a}{\sqrt{2}\sigma_j} \right) - \text{erf}\left( \frac{\mu - \mu_j - a}{\sqrt{2}\sigma_j} \right) \right) \\
&\quad - \sigma_j \left( \mathcal{N}\left( \frac{\mu - \mu_j + a}{\sigma_j}; 0, 1 \right) - \mathcal{N}\left( \frac{\mu - \mu_j - a}{\sigma_j}; 0, 1 \right) \right),
\end{aligned}
\tag{51}
$$

and

$$
\begin{aligned}
&\text{cov}_{p,2}(t^2, \psi_j(t)) = \\
&\int_{\mu-a}^{\mu+a} t^2\mathcal{N}(t; \mu_j, \sigma_j^2) - \frac{1}{2a} \left( \int_{\mu-a}^{\mu+a} t^2 \right) \left( \int_{\mu-a}^{\mu+a} \mathcal{N}(t; \mu_j, \sigma_j^2) \right) \\
&= \int_{\frac{\mu-\mu_j-a}{\sigma_j}}^{\frac{\mu-\mu_j+a}{\sigma_j}} (\mu_j + \sigma_j s)^2 \mathcal{N}(s; 0, 1) - \underbrace{\frac{1}{2a} \left( \frac{(\mu+a)^3}{3} - \frac{(\mu-a)^3}{3} \right)}_{= \frac{a^2}{3} + \mu^2} \left( \int_{\frac{\mu-\mu_j-a}{\sigma_j}}^{\frac{\mu-\mu_j+a}{\sigma_j}} \mathcal{N}(s; 0, 1) \right) \\
&= \left( \mu_j^2 - \mu^2 - \frac{a^2}{3} \right) \int_{\frac{\mu-\mu_j-a}{\sigma_j}}^{\frac{\mu-\mu_j+a}{\sigma_j}} \mathcal{N}(s; 0, 1) + 2\mu_j\sigma_j \int_{\frac{\mu-\mu_j-a}{\sigma_j}}^{\frac{\mu-\mu_j+a}{\sigma_j}} s\mathcal{N}(s; 0, 1) + \sigma_j^2 \int_{\frac{\mu-\mu_j-a}{\sigma_j}}^{\frac{\mu-\mu_j+a}{\sigma_j}} s^2\mathcal{N}(s; 0, 1) \\
&= \left( \mu_j^2 - \mu^2 + \sigma_j^2 - \frac{a^2}{3} \right) \left( \text{erf}\left( \frac{\mu - \mu_j + a}{\sqrt{2}\sigma_j} \right) - \text{erf}\left( \frac{\mu - \mu_j - a}{\sqrt{2}\sigma_j} \right) \right) \\
&\quad - \sigma_j(\mu + \mu_j + a)\mathcal{N}\left( \frac{\mu - \mu_j + a}{\sigma_j}; 0, 1 \right) + \sigma_j(\mu + \mu_j - a)\mathcal{N}\left( \frac{\mu - \mu_j - a}{\sigma_j}; 0, 1 \right).
\end{aligned}
\tag{52}
$$

**G.3  Continuous sparsemax in 2D ($\alpha = 2$, $D = 2$)**

Let us now consider the case where $D = 2$. For $\phi(t) = [t, tt^\top]$, the distribution $p = \hat{p}_{\Omega_2}[f_\theta]$, with $f_\theta(t) = \theta^\top \phi(t)$, becomes a bivariate truncated paraboloid where $\mu$ and $\Sigma$ are related to the canonical parameters as before, $\theta = [\Sigma^{-1}\mu, -\frac{1}{2}\Sigma^{-1}]$. We obtain expressions for the attention mechanism output $\rho_2(\theta) = \mathbb{E}_p[\psi(t)]$ and its Jacobian $J_{\rho_2}(\theta) = \text{cov}_{p,2}(\phi(t), \psi(t))$ that include 1D integrals (simple to integrate numerically), when $\psi(t)$ are Gaussian RBFs, *i.e.*, when each $\psi_j$ is of the form $\psi_j(t) = \mathcal{N}(t; \mu_j, \Sigma_j)$.

We start with the following lemma:

---

**Lemma 1.** *Let $\mathcal{N}(t, \mu, \Sigma)$ be a D-dimensional multivariate Gaussian, Let $A \in \mathbb{R}^{D \times R}$ be a full column rank matrix (with $R \leq D$), and $b \in \mathrm{R}^D$. Then we have $\mathcal{N}(Au + b; \mu, \Sigma) = \tilde{s}\mathcal{N}(u; \tilde{\mu}, \tilde{\Sigma})$ with:*

$$
\begin{aligned}
\tilde{\Sigma} &= (A^\top \Sigma^{-1} A)^{-1} \\
\tilde{\mu} &= \tilde{\Sigma} A^\top \Sigma^{-1} (\mu - b) \\
\tilde{s} &= (2\pi)^{\frac{R-D}{2}} \frac{|\tilde{\Sigma}|^{1/2}}{|\Sigma|^{1/2}} \exp\left(-\frac{1}{2}(\mu - b)^\top P(\mu - b)\right), \quad P = \Sigma^{-1} - \Sigma^{-1} A \tilde{\Sigma} A^\top \Sigma^{-1}.
\end{aligned}
$$

*If $R = D$, then $A$ is invertible and the expressions above can be simplified to:*

$$
\begin{aligned}
\tilde{\Sigma} &= A^{-1} \Sigma A^{-\top} \\
\tilde{\mu} &= A^{-1}(\mu - b) \\
\tilde{s} &= |A|^{-1}.
\end{aligned}
$$

---

*Proof.* The result can be derived by writing $\mathcal{N}(Au + b; \mu, \Sigma) = (2\pi)^{-R/2}|\Sigma|^{-1/2} \exp(-\frac{1}{2}(Au + b - \mu)^\top \Sigma^{-1}(Au + b - \mu))$ and splitting the exponential of the sum as a product of exponentials. □

**Forward pass.** For the forward pass, we need to compute

$$
\mathbb{E}_p[\psi_j(t)] = \iint_{\mathbb{R}^2} \left[ -\lambda - \frac{1}{2}(t - \mu)^\top \Sigma^{-1}(t - \mu) \right]_+ \mathcal{N}(t; \mu_j, \Sigma_j) dt, \tag{53}
$$

with

$$
\mathcal{N}(t; \mu_j, \Sigma_j) = \frac{1}{2\pi |\Sigma_j|^{\frac{1}{2}}} \exp\left(-\frac{1}{2}(t - \mu_j)^\top \Sigma_j^{-1}(t - \mu_j)\right), \tag{54}
$$

and (from (11))

$$
\lambda = -\left(\frac{1}{\pi\sqrt{\det(\Sigma)}}\right)^{\frac{1}{2}}. \tag{55}
$$

Using Lemma 1 and the change of variable formula (which makes the determinants cancel), we can reparametrize $u = (-2\lambda)^{-\frac{1}{2}}\Sigma^{-\frac{1}{2}}(t - \mu)$ and write this as an integral over the unit circle:

$$
\mathbb{E}_p[\psi_j(t)] = \iint_{\|u\| \leq 1} -\lambda(1 - \|u\|^2)\mathcal{N}(u; \tilde{\mu}, \tilde{\Sigma}) du, \tag{56}
$$

with $\tilde{\mu} = (-2\lambda)^{-\frac{1}{2}}\Sigma^{-\frac{1}{2}}(\mu_j - \mu)$, $\tilde{\Sigma} = (-2\lambda)^{-1}\Sigma^{-\frac{1}{2}}\Sigma_j\Sigma^{-\frac{1}{2}}$. We now do a change to polar coordinates, $u = (r\cos\theta, r\sin\theta) = ar$, where $a = [\cos\theta, \sin\theta]^\top \in \mathbb{R}^{2 \times 1}$. The integral becomes:

$$
\begin{aligned}
\mathbb{E}_p[\psi_j(t)] &= \int_0^{2\pi} \int_0^1 -\lambda(1 - r^2)\mathcal{N}(ar; \tilde{\mu}, \tilde{\Sigma})r \, dr \, d\theta \\
&= \int_0^{2\pi} \int_0^1 -\lambda r(1 - r^2)\tilde{s}\mathcal{N}(r; r_0, \sigma^2) \, dr \, d\theta, \tag{57}
\end{aligned}
$$

where in the second line we applied again Lemma 1, resulting in

$$
\begin{aligned}
\sigma^2(\theta) \equiv \sigma^2 &= (a^\top \tilde{\Sigma}^{-1} a)^{-1} \\
r_0(\theta) \equiv r_0 &= \sigma^2 a^\top \tilde{\Sigma}^{-1} \tilde{\mu} \\
\tilde{s}(\theta) \equiv \tilde{s} &= \frac{1}{\sqrt{2\pi}} \frac{\sigma}{|\tilde{\Sigma}|^{1/2}} \exp\left(-\frac{1}{2} \tilde{\mu}^\top P \tilde{\mu}\right), \quad P = \tilde{\Sigma}^{-1} - \sigma^2 \tilde{\Sigma}^{-1} a a^\top \tilde{\Sigma}^{-1}.
\end{aligned}
$$

Applying Fubini's theorem, we fix $\theta$ and integrate with respect to $r$. We use the formulas (50) and the fact that, for any $u, v \in \mathbb{R}$ such that $u \leq v$:

$$
\int_u^v t^3 \mathcal{N}(t; 0, 1) = -\mathcal{N}(v; 0, 1)(2 + v^2) + \mathcal{N}(u; 0, 1)(2 + u^2). \tag{58}
$$

We obtain a closed from expression for the inner integral:

$$
\begin{aligned}
F(\theta) &= \int_0^1 r(1 - r^2) \mathcal{N}(r; r_0, \sigma^2)\, dr \\
&= (2\sigma^3 + r_0^2 \sigma + r_0 \sigma) \mathcal{N}\left(\frac{1 - r_0}{\sigma}; 0, 1\right) - (2\sigma^3 + r_0^2 \sigma - \sigma) \mathcal{N}\left(-\frac{r_0}{\sigma}; 0, 1\right) \\
&\quad - \frac{r_0^3 + (3\sigma^2 - 1)r_0}{2} \left[\operatorname{erf}\left(\frac{1 - r_0}{\sqrt{2}\sigma}\right) - \operatorname{erf}\left(-\frac{r_0}{\sqrt{2}\sigma}\right)\right]. \tag{59}
\end{aligned}
$$

The desired integral can then be expressed in a single dimension as

$$
\mathbb{E}_p[\psi_j(t)] = -\lambda \int_0^{2\pi} \tilde{s}(\theta) F(\theta), \tag{60}
$$

which may be integrated numerically.

**Backward pass.** For the backward pass we need to solve

$$
\operatorname{cov}_{p,2}(t, \psi_j(t)) = \iint_E t \mathcal{N}(t; \mu_j, \Sigma_j) - \frac{1}{|E|} \left(\iint_E t\right) \left(\iint_E \mathcal{N}(t; \mu_j, \Sigma_j)\right) \tag{61}
$$

and

$$
\operatorname{cov}_{p,2}(tt^\top, \psi_j(t)) = \iint_E tt^\top \mathcal{N}(t; \mu_j, \Sigma_j) - \frac{1}{|E|} \left(\iint_E tt^\top\right) \left(\iint_E \mathcal{N}(t; \mu_j, \Sigma_j)\right) \tag{62}
$$

where $E = \operatorname{supp}(p) = \{t \in \mathbb{R}^2 \mid \frac{1}{2}(t - \mu)^\top \Sigma^{-1}(t - \mu) \leq -\lambda\}$ denotes the support of the density $p$, a region bounded by an ellipse. Note that these expressions include integrals of vector-valued functions and that (61) and (62) correspond to the first to second and the third to sixth row of the Jacobian, respectively. The integrals that do not include Gaussians have closed form expressions and can be computed as

$$
\frac{1}{|E|} \left(\iint_E t\right) = \mu \tag{63}
$$

and

$$
\frac{1}{|E|} \left(\iint_E tt^\top\right) = \mu\mu^\top + \frac{\Sigma}{|E|}, \tag{64}
$$

where $|E|$ is the area of the region $E$ given by

$$
|E| = \frac{\pi}{\sqrt{\det\left(\frac{1}{-2\lambda} \Sigma^{-1}\right)}}. \tag{65}
$$

All the other integrals are solved using the same affine transformation and change to polar coordinates as in the forward pass. Given this, $\tilde{\mu}, \tilde{\Sigma}, a, \sigma^2, r_0$ and $\tilde{s}$ are the same as before. To solve (61) we write

$$
\iint_E t \mathcal{N}(t; \mu_j, \Sigma_j) = \iint_{\|u\| \leq 1} \left((-2\lambda)^{\frac{1}{2}} \Sigma^{\frac{1}{2}} u + \mu\right) \mathcal{N}(u; \tilde{\mu}, \tilde{\Sigma}) du \tag{66}
$$

in polar coordinates,

$$\int_0^{2\pi} \int_0^1 r \left( (-2\lambda)^{\frac{1}{2}} \Sigma^{\frac{1}{2}} ar + \mu \right) \tilde{s}\, \mathcal{N}(r; r_0, \sigma^2) dr\, d\theta, \tag{67}$$

which can be then expressed in a single dimension as

$$\iint_E t\mathcal{N}(t; \mu_j, \Sigma_j) \;=\; \int_0^{2\pi} \tilde{s}(\theta) G(\theta) d\theta, \tag{68}$$

with

$$
\begin{aligned}
G(\theta) &= \int_0^1 r \left( (-2\lambda)^{\frac{1}{2}} \Sigma^{\frac{1}{2}} ar + \mu \right) \mathcal{N}(r; r_0, \sigma^2)\, dr \\
&= \int_{-\frac{r_0}{\sigma}}^{\frac{1-r_0}{\sigma}} (s\sigma + r_0) \left( (-2\lambda)^{\frac{1}{2}} \Sigma^{\frac{1}{2}} a(s\sigma + r_0) + \mu \right) \mathcal{N}(r; r_0, \sigma^2)\, ds \\
&= \left( (-2\lambda)^{\frac{1}{2}} \Sigma^{\frac{1}{2}} a\sigma(r_0) + \mu\sigma \right) \mathcal{N}\left( -\frac{r_0}{\sigma}; 0, 1 \right) \\
&\quad - \left( (-2\lambda)^{\frac{1}{2}} \Sigma^{\frac{1}{2}} a\sigma(1 + r_0) + \mu\sigma \right) \mathcal{N}\left( \frac{1 - r_0}{\sigma}; 0, 1 \right) \\
&\quad + \frac{1}{2} \left( (-2\lambda)^{\frac{1}{2}} \Sigma^{\frac{1}{2}} a(\sigma^2 + r_0^2) + \mu r_0 \right) \left[ \operatorname{erf}\left( \frac{1 - r_0}{\sqrt{2}\sigma} \right) - \operatorname{erf}\left( -\frac{r_0}{\sqrt{2}\sigma} \right) \right]. \tag{69}
\end{aligned}
$$

We do the same for

$$\iint_E \mathcal{N}(t; \mu_j, \Sigma_j) = \iint_{\|u\|\le 1} \mathcal{N}(u; \tilde{\mu}, \tilde{\Sigma}) du = \int_0^{2\pi} \int_0^1 r\tilde{s}\, \mathcal{N}(r; r_0, \sigma^2) dr\, d\theta, \tag{70}$$

which can then be expressed in a single dimension as

$$\iint_E \mathcal{N}(t; \mu_j, \Sigma_j) \;=\; \int_0^{2\pi} \tilde{s}(\theta) H(\theta) d\theta, \tag{71}$$

with

$$
\begin{aligned}
H(\theta) &= \int_0^1 r\mathcal{N}(r; r_0, \sigma^2)\, dr = \int_{-\frac{r_0}{\sigma}}^{\frac{1-r_0}{\sigma}} (s\sigma + r_0)\mathcal{N}(r; r_0, \sigma^2)\, ds \\
&= \sigma \left[ \mathcal{N}\left( -\frac{r_0}{\sigma}; 0, 1 \right) - \mathcal{N}\left( \frac{1 - r_0}{\sigma}; 0, 1 \right) \right] + \frac{r_0}{2} \left[ \operatorname{erf}\left( \frac{1 - r_0}{\sqrt{2}\sigma} \right) - \operatorname{erf}\left( -\frac{r_0}{\sqrt{2}\sigma} \right) \right].
\end{aligned}
$$

Finally, to solve (62) we simplify the integral

$$
\begin{aligned}
\iint_E tt^\top \mathcal{N}(t; \mu_j, \Sigma_j) &= \iint_{\|u\|\le 1} \left( (-2\lambda)^{\frac{1}{2}} \Sigma^{\frac{1}{2}} u + \mu \right) \left( (-2\lambda)^{\frac{1}{2}} \Sigma^{\frac{1}{2}} u + \mu \right)^\top \mathcal{N}(u; \tilde{\mu}, \tilde{\Sigma}) du \\
&= \int_0^{2\pi} \int_0^1 r(r^2 A + rB + C)\tilde{s}\, \mathcal{N}(r; r_0, \sigma^2) dr\, d\theta \tag{72}
\end{aligned}
$$

with

$$A = (-2\lambda)\Sigma^{\frac{1}{2}} aa^\top (\Sigma^{\frac{1}{2}})^\top \tag{73}$$

$$B = (-2\lambda)^{\frac{1}{2}} \left( \Sigma^{\frac{1}{2}} a\mu^\top + \mu a^\top (\Sigma^{\frac{1}{2}})^\top \right) \tag{74}$$

$$C = \mu\mu^\top. \tag{75}$$

The integral can then be expressed in a single dimension as

$$\iint_E tt^\top \mathcal{N}(t; \mu_j, \Sigma_j) \;=\; \int_0^{2\pi} \tilde{s}(\theta) M(\theta) d\theta, \tag{76}$$

with

$$
\begin{aligned}
M(\theta) &= \int_0^1 (r^3 A + r^2 B + rC)\mathcal{N}(r; r_0, \sigma^2)dr \\
&= \int_{-\frac{r_0}{\sigma}}^{\frac{1-r_0}{\sigma}} (s^3 \tilde{A} + s^2 \tilde{B} + s\,\tilde{C} + \tilde{D})\mathcal{N}(s; 0, 1)\,ds \\
&= \left[\left(2 + \left(-\frac{r_0}{\sigma}\right)^2\right)\tilde{A} - \frac{r_0}{\sigma}\tilde{B} + \tilde{C}\right]\mathcal{N}\left(-\frac{r_0}{\sigma}; 0, 1\right) \\
&\quad - \left[\left(2 + \left(\frac{1-r_0}{\sigma}\right)^2\right)\tilde{A} + \frac{1-r_0}{\sigma}\tilde{B} + \tilde{C}\right]\mathcal{N}\left(\frac{1-r_0}{\sigma}; 0, 1\right) \\
&\quad + \frac{1}{2}\left(\tilde{B} + \tilde{D}\right)\left[\mathrm{erf}\left(\frac{1-r_0}{\sqrt{2}\sigma}\right) - \mathrm{erf}\left(-\frac{r_0}{\sqrt{2}\sigma}\right)\right]
\end{aligned}
\tag{77}
$$

where

$$
\tilde{A} = \sigma^3 A
\tag{78}
$$

$$
\tilde{B} = \sigma^2(3r_0\,A + B)
\tag{79}
$$

$$
\tilde{C} = \sigma(3r_0^2\,A + 2r_0\,B + C)
\tag{80}
$$

$$
\tilde{D} = r_0^3\,A + r_0^2\,B + r_0\,C.
\tag{81}
$$

# H  Experimental Details and Model Hyperparameters

## H.1  Document classification

We used the IMDB movie review dataset [29],[6] which consist of user-written text reviews with binary labels (positive/negative). Following [43], we used 25K training documents, 10% of which for validation, and 25K for testing. The training and test sets are perfectly balanced: 12.5K negative and 12.5K positive examples. The documents have 280 words on average.

Our architecture is the same as [29], a BiLSTM with attention. We used pretrained GloVe embeddings from the 840B release,[7] kept frozen. We tuned three hyperparameters using the discrete softmax attention baseline: learning rate within $\{0.003, \mathbf{0.001}, 0.0001\}$; $\ell_2$ within $\{0.01, 0.001, \mathbf{0.0001}, 0\}$; number of epochs within $\{5, \mathbf{10}, 20\}$. We picked the best configuration by doing a grid search and by taking into consideration the accuracy on the validation set (selected values in bold). Table 3 shows the hyperparameters and model configurations used for all document classification experiments.

## H.2  Machine translation

We used the De→En dataset from the IWSLT 2017 evaluation campaign [30], with the standard splits (206K, 9K, and 2K sentence pairs for train/dev/test).[8] We used BPE [52] with 32K merges to reduce the vocabulary size. Our implementation is based on Joey-NMT [53] and we used the provided configuration script for the baseline, a BiLSTM model with discrete softmax attention[9] with the hyperpameters in Table 4.

## H.3  Visual question answering

We used the VQA-v2 dataset [31] with the standard splits (443K, 214K, and 453K question-image pairs for train/dev/test, the latter subdivided into test-dev, test-standard, test-challenge and test-reserve). We adapted the implementation of [32],[10] consisting of a Modular Co-Attention Network

Table 3: Hyperparmeters for document classification.

| HYPERPARAMETER | VALUE |
|---|---|
| Batch size | 16 |
| Word embeddings size | 300 |
| BiLSTM hidden size | 128 |
| Merge BiLSTM states | Concat |
| Attention scorer | [14] |
| Conv filters | 128 |
| Conv kernel size | 3 |
| Early stopping patience | 5 |
| Number of epochs | 10 |
| Optimizer | Adam |
| $\ell_2$ regularization | 0.0001 |
| Learning rate | 0.001 |

Table 4: Hyperparmeters for neural machine translation.

| HYPERPARAMETER | VALUE |
|---|---|
| Batch size | 80 |
| Word embeddings size | 620 |
| BiLSTM hidden size | 1000 |
| Attention scorer | [14] |
| Early stopping patience | 8 |
| Number of epochs | 100 |
| Optimizer | Adam |
| $\ell_2$ regularization | 0 |
| Dropout | 0.0 |
| Hidden dropout | 0.2 |
| Learning rate | 0.0002 |
| Scheduling | Plateau |
| Decrease factor | 0.7 |
| Lower case | True |
| Normalization | Tokens |
| Maximum output length | 80 |
| Beam size | 5 |
| RNN type | GRU |
| RNN layers | 1 |
| Input feeding | True |
| Init. hidden | Bridge |

(MCAN). Our architecture is the same as [32] except that we represent the image input with grid features generated by a ResNet [54] pretrained on ImageNet [55], instead of bounding-box features [56]. The images are resized to $448 \times 448$ before going through the ResNet that outputs a feature map of size $14 \times 14 \times 2048$. To represent the input question words we use 300-dimensional GloVe word embeddings [57], yielding a question feature matrix representation. Table 5 shows the hyperparameters used for all the VQA experiments presented.

All the models we experimented with use the same features and were trained only on the train set without data augmentation.

**Examples.** Figure 4 illustrates the difficulties that continuous attention models may face when trying to focus on objects that are too far from each other or that seem to have different relative importance to answer the question. Intuitively, in VQA, this becomes a problem when counting objects in those conditions. On the other side, in counting questions that require the understanding of a contiguous region of the image only, continuous attention may perform better (see Figure 5).

Figures 6 and 7 show other examples where continuous attention focus on the right region of the image and answers the question correctly. For these cases, discrete attention is more diffuse than its

Table 5: Hyperparmeters for VQA.

| HYPERPARAMETER | VALUE |
| --- | --- |
| Batch size | 64 |
| Word embeddings size | 300 |
| Input image features size | 2048 |
| Input question features size | 512 |
| Fused multimodal features size | 1024 |
| Multi-head attention hidden size | 512 |
| Number of MCA layers | 6 |
| Number of attention heads | 8 |
| Dropout rate | 0.1 |
| MLP size in flatten layers | 512 |
| Optimizer | Adam |
| Base learning rate at epoch $t$ starting from 1 | $\min(2.5t \cdot 10^{-5}, 1 \cdot 10^{-4})$ |
| Learning rate decay ratio at epoch $t \in \{10, 12\}$ | 0.2 |
| Number of epochs | 13 |

Figure 4: Attention maps for an example in VQA-v2: original image, discrete attention, continuous softmax, and continuous sparsemax.

continuous counterpart: in both examples, it attends to two different regions in the image, leading to incorrect answers.

Figure 5: Attention maps for an example in VQA-v2: original image, discrete attention, continuous softmax, and continuous sparsemax.

Figure 6: Attention maps for an example in VQA-v2: original image, discrete attention, continuous softmax, and continuous sparsemax.

Figure 7: Attention maps for an example in VQA-v2: original image, discrete attention, continuous softmax, and continuous sparsemax.

## Footnotes

[6]https://ai.stanford.edu/~amaas/data/sentiment

[7]http://nlp.stanford.edu/data/glove.840B.300d.zip

[8]https://wit3.fbk.eu/mt.php?release=2017-01-trnted

[9]https://github.com/joeynmt/joeynmt/blob/master/configs/iwslt14_deen_bpe.yaml

[10]https://github.com/MILVLG/mcan-vqa