[Reviews · NeurIPS 2020]

Review 1

Summary and Contributions: Recent work has introduced methods to parameterize probability distributions over categories that have varying/sparse support, i.e. distributions that assign zero probability to irrelevant categories. These methods use the sparsemax and entmax transformations, rather than the conventional softmax transformation which always assigns nonzero probability to every class. However, this prior work only applied to probability distributions over discrete spaces. The paper under review builds on these works and extends these results to the continuous-domain setting. In particular they introduce a formalism to generalize the notion of sparse vectors to continuous domains. On the experimental side, the authors perform experiments on text classification, machine translation, and visual question answering.

Strengths: At this stage, the main contributions of this work are mostly theoretical. While prior works have limited attention mechanisms to discrete domains, this paper applies them to continuous domains. The paper establishes a link between two lines of work, in statistical physics and in machine learning, namely the deformed exponential families from statistical physics, and sparse versions of the softmax transformation (entmax and sparsemax).

Weaknesses: The experimental results aren’t very convincing. The results (Table 2) show similar accuracies for all attention models, with a slight advantage for continuous softmax. It would help to have a short algorithm describing how to implement the forward and backward passes efficiently. I can see that appendix G discusses how forward and backward passes can be implemented, but it mixes the implementation with many lines of calculations, so it is not easy to understand how the implementation is done.

Correctness: As far as I could see, all claims seem sound.

Clarity: The paper is very well written and pleasant to read.

Relation to Prior Work: The paper clearly explains prior works and the contributions of this paper with respect to these prior works.

Reproducibility: Yes

Additional Feedback: The beta-escort distribution (defined in footnote 4) ought to be defined by a proper equation in the main text, not in a footnote. I was confused about the meaning of the tilde sign above p in Proposition 2 (equation 7), until I saw footnote 4. This would also avoid referring to footnote 4 at line 194. Line 134: “a condition for such solution to exist is g being strongly convex”. Is this a necessary or a sufficient condition? Minor comments. I found the abstract a bit too technical and not easy to understand at first for a non expert. Lines 70 and 126: the notation Sigma > 0 is not defined. It could help to explain that this notation is used to mean “positive definite”. Lines 148-149: there seems to be a typo. “each piece has as a D-dimensional representation” == post-rebuttal == thank for answering my question, and congratulations on the great work!


Review 2

Summary and Contributions: This paper generalizes regularized prediction map [7] to continuous domains allowing sparse support attention (with 0 attention weights) on underlying continuous domains (instead of set of elements). Their theoretical contribution bridges Machine Learning and Statistical Physics by using $$\beta$$-logarithm and the $$\alpha$$-Tsallis' entropy. The authors also derive the Jacobian of their operator to enable backpropagation in modern differential programming frameworks.

Strengths: The theoretical contribution generalizes further the $$\Omega$$-RPM formulation of sparsemax and $$\alpha$$-entmax introduced by Blondel et al. to continuous domains. The $\alpha=2$ case in the continuous domain is particularly interesting because it leads to sparse support of the attention weights such as the Truncated Parabola which could improve computational efficiency and interpretability of the attention model. The supporting experiments tackle diverse tasks showing applicability of continuous attention to document classification, Neural Machine Translation and Visual Question Answering. This involves both text (1D) and image (2D) data modalities. Continuous attention seems to improve performance of the attention model when used together with discrete attention.

Weaknesses: Defining the (continuous) value function using a multivariate ridge regression seems overcomplicated. Have you experimented with other ways such as linear interpolation (1D) or bilinear (2D) or maybe higher order interpolation. Another concern is that I think that ridge regression is order equivarient so does not consider proximity of tokens/pixels to construct the continuous value function. The experimental results are encouraging but do not exhibit a large improvement. The authors do not oversell their achievements. It is sufficient that continuous attention performs on par other attention schema. It also improves interpretability with sparse support of attention.

Correctness: Yes.

Clarity: Yes. The paper is very well written and gives good references for the machine learning community to build on strong results from statistical physics.

Relation to Prior Work: The authors clearly state previous contribution of $$\alpha$$-entmax and sparsemax. I am not confident to assess if the statistical physics literature is properly cited but the references seem comprehensive. The authors should mention that the discrete Gaussian RBF attention was used in [A] with the vector $$[t, vec(tt^\top)]$$ injected as relative positional encoding. This leads to another question: given that continuous attention defines f on a continuous space, say $$[0, 1]^2$$ for images instead of a set of pixels, does it remove the need for additive positional encoding? I would find continuous attention to be a more principled way to encode positions. [A] On the Relationship between Self-Attention and Convolutional Layers Jean-Baptiste Cordonnier, Andreas Loukas, Martin Jaggi ICLR 2020

Reproducibility: Yes

Additional Feedback: Minor: * it would be helpful to repeat what is $M$ around line 170. It was defined 3 pages earlier at line 74 but not reused in between. * line 239: I find that "100 << 14^2" is a bit exaggerating, there is still a center of Gaussian every ~2 cells of the grid.


Review 3

Summary and Contributions: This paper proposes a new continuous attention mechanism, which is an extension of alpha-entmax. Conventional attention mechanism operates on discrete units. On the other hand, the proposed method enable attention on continuous input signal. The main contribution is the theoretical derivation of continuous attention and its gradient for back-propagation. The idea of continuous attention, to the best of my knowledge, is unexplored before. The authors demonstrate the idea on text classification, neural machine translation, and visual question answering. However, the improvements on the evaluation metrics are marginal. Perhaps the main practical advantage is a smoother attention heatmap that is more interpret-able.

Strengths: The main strength of this paper is the novelty: continuous attention has never been explored before. I think this will be of interest to the community since recently attention-based deep learning models are gaining universal success on all machine learning benchmarks. Moreover, attention mechanism is starting to be applied to the vision domain. I think the continuous attention makes much more sense on image data than text data, and it could potentially help to train better attention-based model for computer vision. Due to memory constraints of current GPU hard ware, CNN models are forced to down-sample or use a stride to obtain smaller discrete feature maps. However, images are naturally continuous. Similar issues are presented in videos (down-sampled in the time axis), too. This method could be one interesting way to apply attention on vision tasks, where computation cost is reduced via a smaller number of basis vectors. Unfortunately recent SOTA on VQA is based on discrete region features. If the authors can demonstrate improvements on other vision tasks, it will strengthen this paper a lot. Nevertheless, the novelty and the theoretical analysis might by themselves be great contributions already. Perhaps another strength is that the proposed continuous attention provides better interpret-ability of the attention mechanism without sacrificing the actual performance of the task.

Weaknesses: The major weakness of the paper is the empirical result. The NLP tasks uses a bi-LSTM baseline but a recent standard should be the Transformer. Moreover, the proposed method is a continuous extension of the attention mechanism so it probably makes more sense to test it on a fully attention-based model. Based on my previous experience on sentiment classification and neural machine translation, the improvement of proposed method is not significant. Also, after reading this paper I still do not find treating naturally discrete text as continuous stream of semantics very well-motivated. For the VQA experiment, the baseline is also too weak. Although the authors utilized a SOTA model MCAN, the feature used to represent the image is weak. In fact, it may be seen as one limitation of the proposed method: it cannot handle discrete region-based features such as BUTD [1], which is widely adopted to many high-level vision tasks. --------------------- Update after rebuttal -------------------------------- The authors addressed my concern about region-based features for VQA. I would encourage the authors' run BUTD on VQA in the revision to make this paper even stronger. [1] Anderson et al., "Bottom-Up and Top-Down Attention for Image Captioning and Visual Question Answering", CVPR 2018

Correctness: I did not find any correctness issue in the paper.

Clarity: This is a theory-heavy paper therefore most content are very technical and math-intense. For the other part, the paper is well written and easy to read. A minor suggestion is to add some motivation for why to move from discrete to continuous attention early in the introduction.

Relation to Prior Work: This paper does not have a related work section. Related works are properly cited as theorem/findings of those works are used in the proof. To the best of my knowledge, this is the first attempt to make the attention mechanism works on continuous signal so no related work section may not be a problem. Perhaps it will help some reader to add a paragraph to compare and contrast the proposed method to conventional and the sparse variant of discrete attention, for example, on the computation complexity, memory consumption, or empirical time cost.

Reproducibility: Yes

Additional Feedback: My rating is mainly based on the weak empirical performance of the proposed method. If the authors can address my questions/concerns in the rebuttal period, I am willing to increase the score. - Why model text as continuous inputs? Text are naturally discrete tokens. - The proposed method assumes attention probability is single mode. Is this a reasonable assumption? - The proposed method's application on VQA is limited to grid feature. However, region-based features has been shown to be much better in terms of accuracy. How can continuous attention close the gap? Maybe a more reasonable baseline of VQA experiment is the recently proposed enhanced grid feature [1]. - It is not clear which attention is changed to continuous attention for the VQA experiments. MCAN has 3 attentions: image self-attention, text self-attention, and image-text cross-attention. - What is the computation cost of continuous attention mechanism? If it is similar toe conventional discrete attention, can it be applied to deep transformer models with multi-head attention (with efficient implementation)? All the NLP experiments are performed on bi-LSTM but the Transformer has been widely adopted. - Is there any motivation or intuition, other than the theoretical property, how the continuous attention can improve the accuracy of downstream tasks? ------------------------------ Update after rebuttal -------------------------------- I appreciate the authors' response. Most of my questions are answered. Therefore, I increase my ratings from 6 to 7. [1] Jiang et al., "In Defense of Grid Features for Visual Question Answering", CVPR 2020


Review 4

Summary and Contributions: This work considers sparse continuous attention mechanisms. Starting from \Omega-regularized prediction maps which takes Boltzmann distribution as its solution using negentropy, this work proceeds using the more general \alpha-Tsallis negentropy as the regularizer, such that when \alpha=2 and the space is continuous, the resulting pdf takes the form of a truncated function. This work considers truncated Gaussians and truncated triangular as two special cases. The Jacobians are analytical hence these can be directly used in backpropagation. As an application, this work considers sparse continuous attention. The continuous scores are defined using basis functions, and the discrete data (sentences, images) is made continuous via ridge regression. Experiments show that sparse continuous attention reaches competitive performance to discrete attentions (softmax) or sparse discrete attentions (sparsemax).

Strengths: 1. This work is an interesting generalization of sparse discrete attention to sparse continuous attention. 2. Empirically sparse continuous attention reaches comparable performance to existing attention mechanisms, while allowing for selecting compact regions.

Weaknesses: 1. The benefit of continuous attention compared to discrete attention is not clear. Images and sentences are stored as discrete in computers, and especially for sentences, I don't see why continuous attention is better. Can you elaborate on your computational complexity/efficiency claim due to using fix-dimensional basis functions? Or can you get rid of the positional encodings in machine translation since unimodal distributions can naturally take that into account? 2. While this work proposes a unifying framework on how to obtain different forms of sparse continuous distributions, distributions like truncated Gaussians have been around for a long time. ===post-rebuttal=== It's still not very clear to me why continuous attention would be beneficial to NLP applications. I think it'd be nice if the application of this method on very long documents (with fixed number of basis functions) can be further elaborated. Besides, another weakness might be that the sparse support needs to be oval-shaped (for the distributions considered here), unlike the flexibility in discrete cases.

Correctness: Yes.

Clarity: Yes.

Relation to Prior Work: Yes.

Reproducibility: Yes

Additional Feedback: Please see questions raised in weaknesses.

[Author Response · NeurIPS 2020]

**Reviewer 1**

> *"The results (Table 2) show similar accuracies for all attention models."*

Note that the VQA results in Table 2 with continuous attention use fewer basis functions than discrete regions. Although the accuracies are similar, the unimodal attention suggests better interpretability (as noted by R2 and R3).

> *"It would help to have a short algorithm describing how to implement the forward and backward passes efficiently."*

Good idea, we will add this to the camera-ready version.

> *"Line 134: "a condition (...) is g being strongly convex". Is this a necessary or a sufficient condition?"*

Sufficient; we will clarify and follow the suggestions (move the beta-escort definition to the main text and fix typos).

**Reviewer 2**

Thanks for the positive comments and for pointing out the work of Cordonnier et al. (2020). We will add a citation.

> *"Have you experimented with other ways such as linear interpolation (1D) or bilinear (2D)?"*

We chose ridge regression as it enables a closed-form solution expressed linearly in terms of the basis functions (Eq. 15) and matrix $G$ can be precomputed, leading to a fast implementation. Note that the proximity of tokens/pixels is taken into account (the basis vectors $\psi(t_\ell)$ forming $F$ are located at each token/pixel). We haven't tried linear interpolation, but this is an interesting suggestion (although it might make attention computation more challenging).

> *"Does it remove the need for additive positional encoding?"*

Very good point; this is indeed one advantage of our approach – by converting the input to a function on a predefined continuous space, it encodes "positions" implicitly in a natural way, not requiring explicit positional encoding.

**Reviewer 3**

> *"The proposed method's application on VQA is limited to grid feature."*

Actually, our method can handle BUTD features too: it suffices to let the $t_\ell$ coordinates in the multivariate regression (Eq. 15) be placed on these regions instead of on a grid. However, we opted not to rely on an external object detector, in order to check if continuous attention has the ability to detect relevant objects on its own (see ellipses in Fig. 3). However, for a high-level vision system, combining our method with BUTD is an interesting idea.

> *"Why model text as continuous inputs? Text are naturally discrete tokens."*

We agree text is fundamentally a discrete sequence of symbols. However, when processing long documents or attending to snippets, modelling it as a continuous signal may be advantageous, due to smoothness and independence of length.

> *"The proposed method assumes attention probability is single mode. Is this a reasonable assumption?"*

Good point. Unimodal attention is useful to focus on a single object or text segment of varying size, avoiding "fragmenting" attention probability; however, in some applications, multimodal attention may be preferable. Our method can be extended to multiple modes via a suitable choice of $\phi(t)$ (e.g., a mixture of Gaussians), but this will require numeric integration for attention computation. A simpler strategy (see lines 258-260) is to use multi-head or sequential attention.

> *"Can it be applied to deep transformer models with multi-head attention?"*

Great question. We have ongoing work applying this to transformer models (but out of scope for this paper). Briefly, the computation cost is $O(N)$ for each attention head (against $O(L)$ in the discrete case) where $N \ll L$ is the number of basis functions, plus an extra $O(NL)$ cost in the first layer to perform the multivariate regression on $L$ tokens.

> *"Is there any intuition (...) how the continuous attention can improve the accuracy of downstream tasks?"*

In general, continuous attention can make it easier to attend to large spaces with different resolution levels, with a fixed number of Gaussian RBFs with several variances. It can also lead to more focused attention (the VQA experiments suggest this) and better control of time steps with continuous data streams (*e.g.*, irregularly sampled time series). We haven't explored all these directions, but we believe these are promising areas of future research.

**Reviewer 4**

Please check our answers to R1 and R3 above (the answer is yes to positional encodings).

[Meta-Review · NeurIPS 2020]

The reviewers found the paper well written and novel. I would ask the authors to add discussion of previous work on "parameterized" attention, e.g. https://arxiv.org/abs/1502.04623 or https://arxiv.org/abs/1502.04623. (This is not a suggestion that the authors' work is not novel, but rather that there is some context that would be good to include)